# A Set of Generalized Components to Achieve Effective Poison-only Clean-label Backdoor Attacks with Collaborative Sample Selection and Triggers

**Zhixiao Wu**
Harbin Institute of
Technology, Shenzhen
wzxnh24428@gmail.com

**Yao Lu**[*]
Harbin Institute of
Technology, Shenzhen
luyao2021@hit.edu.cn

**Jie Wen**
Harbin Institute of
Technology, Shenzhen
jiewen_pr@126.com

**Hao Sun**
Harbin Institute of
Technology, Shenzhen
hitsz.sh@gmail.com

**Qi Zhou**
Harbin Institute of
Technology, Shenzhen
mickyseveneleven@gmail.com

**Guangming Lu**
Harbin Institute of
Technology, Shenzhen
luguangm@hit.edu.cn

## Abstract

**P**oison-only **C**lean-label **B**ackdoor **A**ttacks (**PCBAs**) aim to covertly inject attacker-desired behavior into DNNs by merely poisoning the dataset without changing the labels. To effectively implant a backdoor, multiple **triggers** are proposed for various attack requirements of **A**ttack **S**uccess **R**ate (**ASR**) and stealthiness. Additionally, **sample selection** enhances clean-label backdoor attacks' ASR by meticulously selecting "hard" samples instead of random samples to poison. Current methods, however, **1)** usually handle the sample selection and triggers in isolation, leading to limited performance on both ASR and stealthiness when converted to PCBAs. Therefore, *we seek to explore the bi-directional collaborative relations between the sample selection and triggers to address the above dilemma.* **2)** Since the strong specificity within triggers, the simple combination of sample selection and triggers fails to flexibly and generally mitigate the drawback of various backdoor attacks. Therefore, *we seek to propose a set of components based on the commonalities of attacks*. Specifically, **Component A** ascertains two critical selection factors, and then makes them an appropriate combination based on the trigger scale to select more reasonable "hard" samples for improving ASR. **Component B** is proposed to select samples with similarities to relevant trigger implanted samples to promote stealthiness. **Component C** reassigns trigger poisoning intensity on RGB colors through distinct sensitivity of the human visual system to RGB for higher ASR, with stealthiness ensured by sample selection including Component B. Furthermore, **all components can be strategically integrated into diverse PCBAs, enabling tailored solutions that balance ASR and stealthiness enhancement for specific attack requirements**. Extensive experiments demonstrate the superiority of our components in stealthiness, ASR, and generalization. Our code can be seen at https://github.com/HITSZ-wzx/GeneralComponents.git.

## 1 Introduction & Related Works

Since the interpretation of **D**eep **N**eural **N**etworks (DNNs) is still under-explored, effectively defending the backdoor attacks (Doan et al. [2021], Lv et al. [2023], Zeng et al. [2023a]) is a huge challenge.

---

[*]Corresponding author

39th Conference on Neural Information Processing Systems (NeurIPS 2025).

Among various types of attacks, **P**oison-only **B**ackdoor **A**ttacks (PBAs) are straightforward to execute since they simply involve contaminating the training dataset for DNNs by embedding pre-designed triggers into selected samples. Such poisoned models will exhibit attacker-desired behavior when processing triggers implanted samples but retain the fundamental functionality with benign samples. Furthermore, **C**lean-label **B**ackdoor **A**ttacks (**CBAs**) (Huynh et al. [2024], Zhao et al. [2024]), preserving the ground-truth sample labels, attract volume attention due to their better resilience against manual inspection. The above settings correspondingly increase the difficulty of designing effective backdoor attacks. Therefore, it is a significant issue to explore an elegant and effective approach to optimize backdoor attacks as effective PCBAs for better applicability and stealthiness.

**Sample selection** (Hayase and Oh [2022], Li et al. [2023d], Li et al. [2024b], Hung-Quang et al. [2024], Wang et al. [2025]) are proposed to enhance ASR by poisoning selected "hard" samples instead of random samples. Therefore, the poisoned models tend to learn the implicit projection between the trigger feature and the target label to evade the difficulty of the original classification upon such samples. The SOTA metric, Forgetting Event, selects "hard" samples by comparing the frequency of misclassification transitions during the pre-training phase. However, the Forgetting Event metric neglects the category information in misclassification transitions, limiting the search of "harder" samples. ***Therefore, it is vital to introduce an appropriate way to employ category information and jointly integrate with Forgetting Event for further optimizing the selection of "hard" samples.*** Furthermore, existing methods neglect the effect of sample selection on the stealthiness enhancement of backdoor attacks. ***Therefore, it is critical to explore an effective mechanism for sample selection on the satisfactory enhancement of stealthiness.***

**Multiple triggers** are designed to effectively implant a backdoor for various attack requirements, which can be mainly classified into three categories. *(1) Invisible triggers characterized by global low-intensity poisoning* are designed for rigorous stealthiness constraints. However, current invisible attacks confront challenges in achieving invisibility with satisfactory ASR in a straightforward way. For instance, BppAttack (Wang et al. [2022]), a stealthy attack based on image quantization and dithering, employs adversarial training and label flipping to embed the low-intensity triggers for ensuring ASR. Therefore, *(2) visible triggers characterized by local high-intensity poisoning* (e.g., Badnets) and *(3) visible triggers characterized by global medium-intensity poisoning* (e.g., Blend) remain valuable for their high deployability and higher ASR. However, the stealthiness of such visible triggers is unsatisfactory. In summary, visible and invisible attacks possess their irreplaceable strengths in different application scenarios. ***Therefore, the optimization method for effective PCBAs should consider the generalization upon various types of triggers in a flexible approach.***

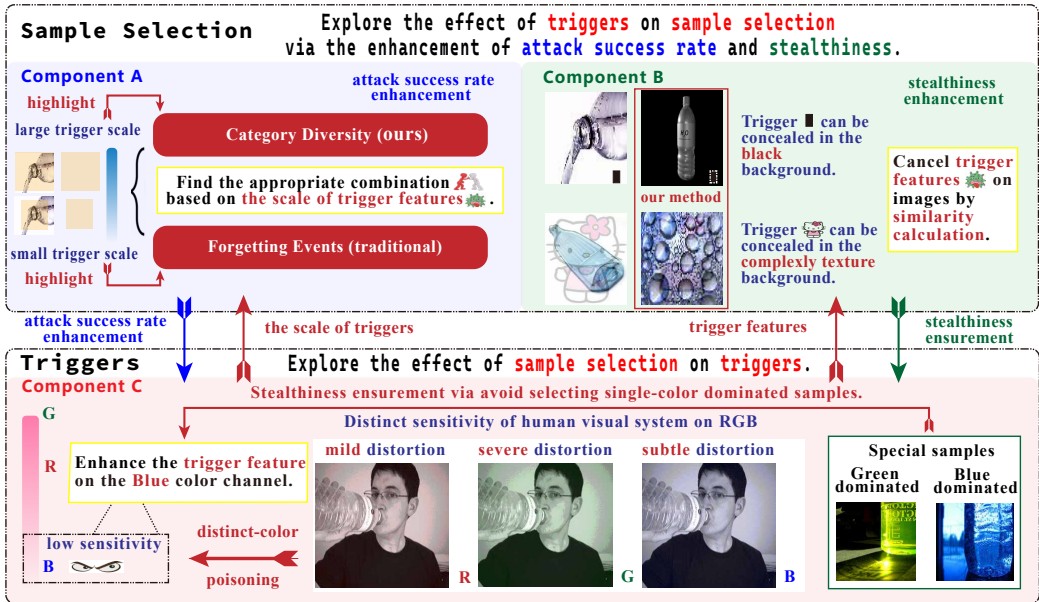

Figure 1: PCBAs optimization by components with collaborative sample selection and triggers.

To resolve the above issues, we propose a set of generalized components, which sufficiently induce the bi-directional collaborative relations between the sample selection and triggers, to significantly

improve both stealthiness and ASR while ensuring generalization for various attacks. Components are demonstrated as below, and more details about related works can be seen at **Appendix A**.

**Component A** ascertains two critical selection factors, and searches for an appropriate combination based on the trigger scale to select more reasonable "hard" samples for improving the ASR of PCBAs. (a) Firstly, we observe the significance of category information on ASR improvement through exploration experiments at **Section 2.1**. Thus, a novel selection factor, Category Diversity, is introduced into sample selection. (b) Secondly, experiments demonstrate that the trigger scales can guide the appropriate combination between Forgetting Event and Category Diversity for selecting more reasonable "hard" samples. Details can be seen at **Section 3.2**.

**Component B** selects samples with similarities to relevant trigger implanted samples via similarity calculation based on appropriate metrics (e.g., Gradient Magnitude Similarity Deviation (Xue et al. [2013])), thereby promoting stealthiness by exploiting the distinct visibility between the human vision system and computer system. Specifically, the trigger feature can be invisible to the human vision system when placed with a similar feature in benign images while maintaining the visibility in views of the computer system, as depicted in **Section 2.2**.

**Component C** is a general optimization on trigger design for higher ASR. (a) Through exploration research at **Appendix E**, we notice the potential of the distinct sensitivity of the human visual system to RGB colors in trigger design. Specifically, poisoning at the blue channel exhibits better stealthiness than poisoning at other colors. Therefore, we reassign trigger poisoning intensity in RGB for a better balance of ASR and stealthiness. (b) Component B prevents the adversary from implanting triggers into blue-dominated samples, thereby further ensuring the stealthiness of enhanced triggers.

In summary, Components A&B introduce the trigger to optimize the sample selection. Specifically, Component A selects more "harder" samples by searching the appropriate combination between Forgetting Event and Category Diversity based on the trigger scale for ASR enhancement. Component B considers the trigger feature to select samples for stealthiness enhancement. Furthermore, Component C reassigns trigger poisoning intensity in RGB for ASR enhancement, of which stealthiness can be ensured by introducing the sample selection, especially Component B. What is more, multiple collaborative components will be effective for different attacks due to the attack commonalities introduced in the mechanism of the above components. Extensive experiments validate the superiority of our components in terms of generalization capability.

## 2  Our Methods

### 2.1  Component A: Appropriate Combination of Metrics Based on Trigger Scale

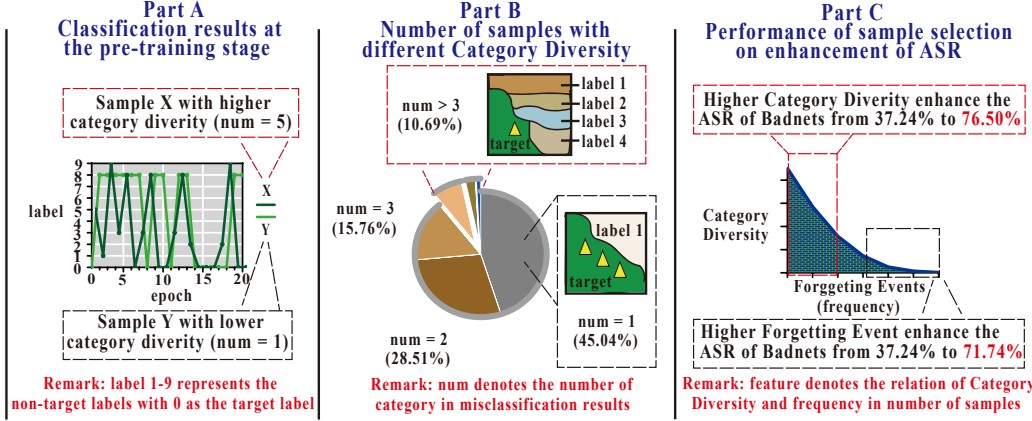

Figure 2: Pilot experiments of Category Diversity. In Part A&B, we explore the significant difference in Category Diversity between samples. In Part C, we ascertain two critical selection factors and the potential internal conflict between Forgetting Event and Category Diversity.

Poisoning the "hard" samples leads the model to learn the strong correspondence between triggers and the target label $y_t$ to avoid the hard-to-learn challenge in such samples. Component A ascertains two critical selection factors and utilizes the *trigger scale* to guide an optimal combination for selecting "harder" samples, thereby enhancing attacks' ASR.

**Forgetting Event** Given a sample $(x_i, y_t)$ in the target-label set $D_t$, Forgetting Event denotes the event when the sample is classified from $y_t$ to $y_m (y_m \neq y_t)$, whose frequency can be represented as $Num_{forget}(x_i)$. Sample selection based on Forgetting Event can be represented as:

$$D_s = arg \max_{D_s \subset D_t} \sum_{(x_i, y_i) \in D_s} Num_{forget}(x_i). \tag{1}$$

**Category Diversity** Through exploration experiments in Figure 2 A&B, samples exhibit a significant difference in category diversity during misclassification events of samples. According to Part C, higher category diversity can enhance the ASR of Badnets, thereby serving as a significant metric in sample selection. We use $\mu$ to represent the mean of $\{Ne((x_i, y_i), y_m)(y_m \neq y_t)\}$. The selected samples are expected to exhibit higher Category Diversity in relatively balanced proportions:

$$D_s = arg \min_{D_s \subset D_t} \sum_{(x_i, y_i) \in D_s} ||N_e((x_i, y_i), y_m) - \mu||_2. \tag{2}$$

We devise a series of distinct negative functions $N_F$ to adjust weights of categories according to the Forgetting Event (frequency) at varied rates $(O(\log(x)), O(x), O(x^2)),$ and $O(e^x))$ for exploring the reasonable combination. Higher rates highlight the significance of Category Diversity in sample selection. We exhibit the details of metric calculation with $N_F$ at $\log(x)$, dubbed 'Res-x', in Algorithm 1 and algorithms with other negative functions in **Appendix C**. Experiments in **Section 3.2** imply that the appropriate combination of two factors depends on the trigger scale.

---

**Algorithm 1** Metric Calculation with Negative Function $N_F$ at $O(\log(x))$

---

**Input :** Train Dataset $D_{tr}$, Target Label $y_t$, Misclassification Events $N_e((x_i, y_i), y_m)$
**Output :** Calculated Metric of Samples
**for** image $(x_i, y_t) \in D_{tr}$ **do**
   $Num[y_m] = 0$
   **for** $y_m \in Y$ **do**
      $Num[y_m] = Num[y_m] + N_e((x_i, y_t), y_m)$
   **end for**
**end for**
**for** $y_m \in Y$ **do**
   $Sum = Sum + log(1 + Num[y_m])$
**end for**
**for** $y_m \in Y$ **do**
   $Cls[y_m] = 1 - \frac{log(1 + Num[y_m])}{Sum}$
**end for**
**for** image $(x_i, y_t) \in D_{tr}$ **do**
   $Metric[x_i] = 0$
   **for** $y_m \in Y$ **do**
      $Metric[x_i] = Metric[x_i] + Cls[y_m] * N_e((x_i, y_t), y_m)$
   **end for**
**end for**

---

## 2.2 Component B: Selection of Samples Exhibiting Visual Insensitivity to Triggers

Typical traditional triggers can be classified as high-intensity local visible triggers (e.g., Badnets), medium-intensity global visible triggers (e.g., Blended), and low-intensity global invisible triggers (e.g., BppAttack). As depicted in Figure 1, Component B enhances the stealthiness of attacks by concealing visible triggers in similar parts of the selected benign images. For example, the $\{h \times w\}$ patch on poisoned images $X_{h \times w}^p$ implanted by Badnets trigger (all-black patch) and the selected images $X_{h \times w}^b$ with $|x_{i,j}| < \epsilon$ can be represented as:

$$X_{h \times w}^p = \begin{bmatrix} 0 & 0 & \dots & 0 \\ 0 & 0 & \dots & 0 \\ \vdots & \vdots & \ddots & \vdots \\ 0 & 0 & \dots & 0 \end{bmatrix}_{h \times w} , \quad X_{h \times w}^b = \begin{bmatrix} x_{0,0} & x_{0,1} & \dots & x_{0,w} \\ x_{1,0} & x_{1,1} & \dots & x_{1,w} \\ \vdots & \vdots & \ddots & \vdots \\ x_{h,0} & x_{h,1} & \dots & x_{h,w} \end{bmatrix}_{h \times w} . \tag{3}$$

Therefore, the trigger can be stealthy on the human vision system when $\epsilon$ is a relatively small integer closest to zero. Furthermore, the strong discriminative property of the all-zero feature in machine learning leads the Badnets trigger to remain visible on the computer system, thereby avoiding the significant decline in ASR. In our paper, we search the samples exhibiting the most visual insensitivity for the Badnets trigger by calculating the sum of **M**ean **S**quared **E**rrors (MSE) in varying patches.

However, Component B with MSE used in global-poisoning triggers (e.g., Blended and BppAttack) is insensitive to severe local distortions. In global poisoning attacks, **G**radient **M**agnitude **S**imilarity **D**eviation (GMSD) shows the superiority in searching appropriate samples for trigger concealment. Details of GMSD can be seen in **Appendix D**. What is more, when collaborating with Component A, the pseudocode for sample selection upon GMSD can be seen in Algorithm 2.

---

**Algorithm 2** Sample Selection with *Components A&B*

---

    **Input :** Target Label $y_t$, Samples selected by *Component A* $D_a$,
    **Input :** The weight of *Component A* $\alpha_s$
    **Output :** Samples selected by *Component A&B* $D_{ab}$
    **Initialize :** Empty array $R_a$ to save the GMSDs of $D_a$
    **for** image $(x_i, y_t) \in D_a$ **do**
        Implant the trigger into image $x_i$ to get poisoned image $x_p$
        Compute the GMSD between image $x_i$ and poisoned image $x_p$
        Save the tuple $[x_i, \text{GMSD}(x_i, x_p)]$ into $R_a$
    **end for**
    sorted_tuples = sorted($R_a$, key=lambda $g$:$g[1]$)
    $D_{ab}$ = [$g[0]$ for $g$ in sorted_tuples[: $\alpha * num(D_a)$]]

---

## 2.3 Component C: Trigger Optimized with Stealthiness Assurance in Sample Selection

In this section, with the stealthiness enhancement by Component B, we further optimize the triggers via distinct-color poisoning for ASR enhancement based on the distinct insensitivity of human visual systems to colors. Research about the human visual system can be seen in **Appendix E**.

**Optimization on Badnets&Blended triggers**    Badnets implant triggers by completely replacing the pixels of the predetermined patch in the original image. According to experiments in **Appendix F**, triggers with alternating black-and-white patterns achieve significantly higher ASR than using single-color patterns. However, black-and-white triggers are more easily detectable by human inspection and pose difficulties in Component B to select images for stealthiness enhancement. Distinct-color poisoning is introduced to combine the advantages of single-color and black-and-white triggers by employing {single-color trigger, single-color trigger, black-and-white trigger} in RGB channels.

For Blended attacks, triggers are linearly blended with the image using specified weighting proportions (e.g., 0.2 in all channels). Experiments in **Section 3.1** demonstrate concentrating the poisoning intensity on a single channel leads to higher ASR compared to the even poisoning way. Distinct-color poisoning reassigns the weight in RGB from $\{0.2, 0.2, 0.2\}$ to $\{0.2, 0.1, 0.3\}$. Therefore, the intensity of triggers in the green channel, which is visually sensitive to humans, is weakened. The enhanced intensity in the blue channel, which is visually insensitive to humans, leads to ASR enhancement.

**Optimization on BppAttack triggers**    BppAttack reduces the color palette of depth from $m_b$ bits to a smaller color depth ($m_p$ bits) in all color channels. The trigger $f_t$ is defined in Eqn.4, where $round$ represents the integer rounding function:

$$f_t(x) = \frac{round(\frac{x}{2^{m_b}-1} * (2^{m_p} - 1))}{2^{m_p} - 1} * (2^{m_b} - 1). \tag{4}$$

The optimized BppAttack, dubbed MultiBpp, optimizes the original quantization process by exploiting the difference of the human visual system to colors. (1) We replace the color palette $m_b, m_p$ in Eqn.4 by the number of representable colors N ($N_b = 2^{m_b} - 1, N_p = 2^{m_p} - 1$) to precisely control the strength of the poisonous feature in Eqn.5. (2) We differentiate the poisoning intensity in the three color channels (e.g., $N_b^c, N_p^c, c \in \{R, G, B\}$ ) instead of maintaining a uniform intensity.

$$\tilde{f}_t^c(x) = \frac{round(\frac{x^c}{N_b^c} * (N_p^c))}{N_p^c} * (N_b^c). \tag{5}$$

We also follow the BppAttack by introducing the Floyd-Steinberg dithering to enhance the stealthiness of the MultiBpp triggers, as depicted in Algorithm 3. We devise two corresponding MultiBpp triggers. One involves poisoning exclusively the blue channel (MultiBpp-B), whereas the other implements differential poisoning across all channels (MultiBpp-RGB).

---

**Algorithm 3** Quantization with Floyd-Steinberg Dithering

**Input :** Selected Samples to be Poisoned $D_s$, Diffusion Distribution $[d_1^c, d_2^c, d_3^c, d_4^c]$
**Output :** Poisoned Samples
**for** image $x \in D_s$ **do**
  **for** $c \in \{R, G, B\}$ **do**
    **for** $i$ from right to left **do**
      **for** $j$ from top to bottom **do**
        $res^c = \tilde{f}_t^c(x^c[i][j]) - x^c[i][j]$
        $x^c[i][j] = x^c[i][j] + res^c$
        $x^c[i+1][j] = x^c[i][j] + res^c * d_1^c$
        $x^c[i+1][j+1] = x^c[i][j] + res^c * d_2^c$
        $x^c[i][j+1] = x^c[i][j] + res^c * d_3^c$
        $x^c[i-1][j+1] = x^c[i][j] + res^c * d_4^c$
      **end for**
    **end for**
  **end for**
**end for**

---

## 3 Experiments

We optimize {Badnets-C, Blended-C, BppAttack} to demonstrate the superiority of our components on various types of attacks {local high-intensity poisoning attacks, global medium-intensity poisoning attacks, global low-intensity poisoning attacks}. Blended-C and Badnets-C represent the clean-label variants of Blended and Badnets attacks. $N_p^R : N_p^G : N_p^B$ in MultiBpp attacks represents the concrete quantization setting of poisoning intensity in RGB channels. Specifically, the default bit depth of BppAttack in the original work is 5, which can be seen as $32 : 32 : 32$ in this paper. Base represents the quantization attack by $32 : 32 : 32$ without the training control and label flipping in BppAttack. Details of attack setup can be seen in **Appendix G**.

### 3.1 Main Results

Table 1: Performance of sample selection upon CIFAR-10 with 1% samples poisoned.

| Sample Selection | | | Badnets-C | | Blended-C | | MultiBpp-B | | MultiBpp-RGB | |
|---|---|---|---|---|---|---|---|---|---|---|
| Type | no. | Selection | ASR | BA | ASR | BA | ASR | BA | ASR | BA |
| Bench | a | Random | 37.24 | 94.42 | 53.41 | 94.90 | **1.37** | 94.51 | **1.16** | 94.95 |
| | b | Loss | 52.71 | 94.71 | 59.43 | **95.10** | 28.02 | 94.84 | 47.85 | 94.76 |
| | c | Gradient | 52.56 | 94.45 | 58.45 | 94.77 | 38.26 | **95.04** | 53.28 | **95.03** |
| | d | Forget | 71.74 | 94.90 | 71.05 | 94.55 | **74.39** | 94.92 | **78.10** | 94.90 |
| Ours | e | Res-$log$ | 82.13 | **94.98** | 82.34 | 94.73 | 77.10 | 94.54 | 80.20 | 94.82 |
| | f | Res-$x$ | 68.65 | 94.71 | 82.31 | 94.31 | 76.73 | 94.21 | 83.07 | 94.63 |
| | g | Res-$x^2$ | 78.76 | 94.94 | **84.88** | 94.38 | **82.54** | 94.58 | **83.88** | 94.59 |
| | h | Res-$e^x$ | 76.50 | 94.47 | 71.81 | **94.80** | 53.92 | **94.72** | 62.28 | **94.85** |

**Effect of Component A with different Negative Functions upon ASR enhancement :** We adopt the same experimental setup as BppAttack (Wang et al. [2022]). We use Res-X ($X \in \{log, x, x^2, e^x\}$) to represent Component A with different Negative Functions in various rates X. According to Table 1, Component A (Ours) significantly enhances the ASR. Specifically, for BadNets attacks, Res-$log$ outperforms the Forget metric (Forgetting Event) upon ASR from $71.74\%$ to $82.13\%$. For Blended attacks, Res-$x^2$ exceeds the Forget metric upon ASR from $71.05\%$ to $84.88\%$. The optimal metrics of {Badnets-C, Blended-C, MultiBpp-B, MultiBpp-RGB} are {Res-$log$, Res-$X$, Res-$x^2$, Res-$x^2$}. Therefore, the significance of Category Diversity is highlighted in global-poisoning attacks. **The ASR decreases on Res-$e^x$ imply the negative impact of inappropriate combination**. The stable ASR enhancement on the {local high-intensity poisoning attack, global medium-intensity poisoning attack, global low-intensity poisoning attack} shows the superiority of Component A on generalization with collaborative sample selection and triggers.

Table 2: Performance of sample selection upon CIFAR-100.

| Sample Selection | | | Poisoning Rate $\alpha = 0.2\%$ | | | | Poisoning Rate $\alpha = 0.5\%$ | | | |
| | | | Badnets-C | | Blended-C | | Badnets-C | | Blended-C | |
| Type | no. | Selection | ASR | BA | ASR | BA | ASR | BA | ASR | BA |
|---|---|---|---|---|---|---|---|---|---|---|
| Bench | a | Random | 7.49 | 77.86 | 40.48 | 77.70 | 51.49 | 78.46 | 65.55 | **78.51** |
| | b | Loss | 17.84 | 78.01 | 46.59 | **78.83** | 70.97 | 78.06 | 70.42 | 78.27 |
| | c | Gradient | 25.25 | **78.28** | 53.03 | 78.62 | **82.02** | **78.67** | 72.51 | 78.33 |
| | d | Forget | **59.39** | 78.21 | **63.11** | 78.10 | 79.69 | 78.53 | **73.31** | 78.43 |
| Ours | e | Res-$log$ | 62.64 | 78.22 | 67.53 | 78.04 | 83.71 | 78.33 | 73.63 | 78.17 |
| | f | Res-$x$ | **80.48** | **78.25** | **73.48** | **78.29** | 84.05 | 78.46 | 76.36 | 78.30 |
| | g | Res-$x^2$ | 72.48 | 78.08 | 66.27 | 78.08 | **85.06** | **78.56** | **77.45** | 77.92 |
| | h | Res-$e^x$ | 52.14 | 78.18 | 60.04 | 78.16 | 71.41 | 78.27 | 72.20 | **78.31** |

The poisoning rate of clean-label attacks is merely $1\%$ in CIFAR-100 when poisoning all target-label samples. Therefore, selecting $50\%$ of the target-label set is reasonable to ensure ASRs of attacks, in which the gap between selecting methods is significantly narrowed. Specifically, according to Table 2, the ASR difference between **Forget** and **Loss** on Badnets-C decreases from $31.55\%$ ($59.39\% - 17.84\%$) to $8.72\%$ ($79.69\% - 70.97\%$). In contrast, our method can still maintain its superiority under relatively larger poisoning rates. Furthermore, Badnets-C achieves $80.48\%$ ASR with Res-$x$ strategy, $21\%$ **higher** than the Forgetting Event metric. Blended-C achieves $73.48\%$ ASR with Res-$x$ strategy, $10\%$ **higher** than Forgetting Event. The optimal methods of Badnets-C on {CIFAR-10, CIFAR-100} are {Res-$log$, Res-$x$}, which indicates the enhanced significance of Category Diversity in datasets with more categories. **In summary, Component A can significantly improve ASR with generalization upon various attacks via searching the appropriate combination of Forgetting Event and Category Diversity based on the trigger scale.**

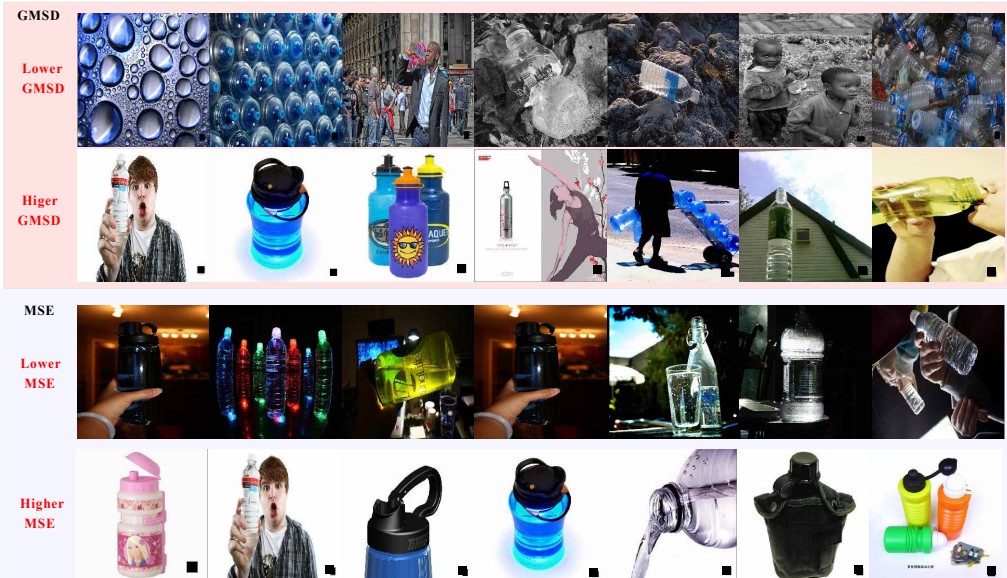

Figure 3: Images poisoned by Badnets attacks with different evaluation metrics in Component B.

**Effect of Component B on stealthiness enhancement:** As depicted in Figure 3, poisoned images with lower GMSD and MSE values both exhibit superiority in the stealthiness of triggers for Badnets attacks. Component B with GMSD tends to find samples with complex colors where the visual sensitivity to MultiBpp triggers will significantly weaken (GMSD $\in [0.0274, 0.0769]$). In contrast, single-color dominated images where GMSD $\in [0.3806, 0.4927]$ are selected by Component B as bad samples. Component B with GMSD tends to find samples with complex backgrounds where the visual sensitivity to Badnets triggers will weaken. In contrast, Component B with MSE tends to find samples with patches similar to the triggers, where the visual sensitivity to Badnets triggers will significantly weaken. Therefore, MSE exhibits superiority in the stealthiness enhancement of Badnets attacks compared to GMSD and will be applied in this paper. Visual performance on {Blend-C, BppAttack} can be seen in **Appendix I**.

We select a specific image set to be visualized according to the stealthiness rankings sorted by Component B for various attacks in Figure 4. Images from top to bottom represent {MultiBpp-B (255:255:8), Blended-C, Badnets-C} and images depicted from left to right represent the GMSD

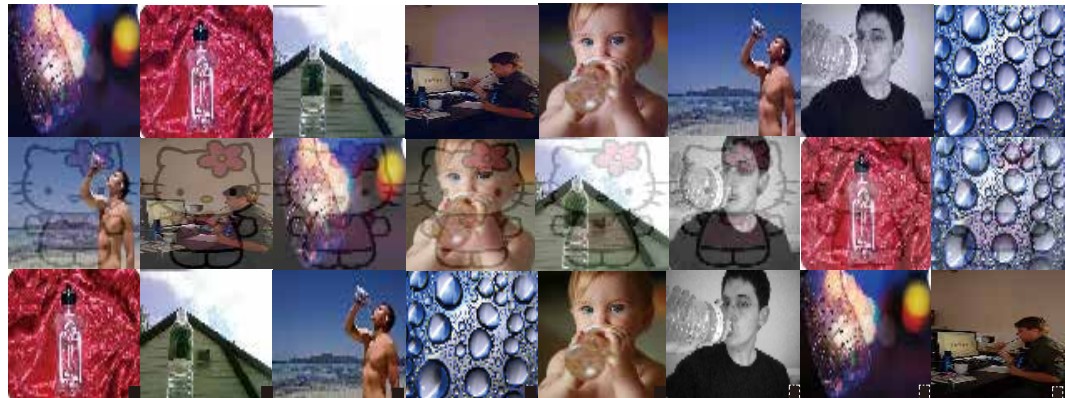

Figure 4: Images sorted by Component B with similarity calculation.

value from high to low, corresponding to the decrease of visual sensitivity measured by Component B. In summary, samples are not equal in visual sensitivity, and the stealthiness ranking of samples dynamically varies in different attacks. In summary, **Component B enhances the stealthiness by concealing the trigger feature on the part of benign images similar to triggers, which can retain generalization ability upon various attacks.** Visual presentation of poisoned images selected by Component B with different MSD and GMSD values is provided in **Appendix I**. For any machine-quantifiable evaluation metric provided, component B can rapidly identify images that achieve optimal performance upon the selected metric.

Table 3: Performance of global-poisoning attacks by poisoning 2.5% samples of CIFAR-10.

| Attack | | | Metric | | Attack Setting | | |
|---|---|---|---|---|---|---|---|
| Type | no. | Method | ASR | BA | Clean-label | Training Control | Stealthy |
| Benchmark | a | Benign | - | **95.0** | ✔ | ✘ | ✔ |
| | b | Base | **8.2** | 94.8 | ✔ | ✘ | ✔ |
| | c | BppAttack | **12.5** | 94.5 | ✘ | ✔ | ✔ |
| | d | Blended-C | **66.4** | 94.3 | ✔ | ✘ | ✘ |
| MultiBpp (our methods) | e | 255:255:8 | 68.6 | 94.8 | ✔ | ✘ | ✔ |
| | f | 255:255:12 | 60.0 | **94.9** | ✔ | ✘ | ✔ |
| | g | 24:48:8 | **76.6** | 94.7 | ✔ | ✘ | ✔ |
| | h | 36:72:12 | 57.7 | 94.6 | ✔ | ✘ | ✔ |
| MultiBpp (others) | i | 8:255:255 | **84.1** | **94.7** | ✔ | ✘ | ✘ |
| | j | 255:8:255 | 72.2 | 94.3 | ✔ | ✘ | ✘ |
| | k | 12:255:255 | 67.6 | 94.5 | ✔ | ✘ | ✘ |
| | l | 255:12:255 | 73.8 | 94.5 | ✔ | ✘ | ✘ |

**Effect of Component C on ASR enhancement:** According to Tables 3c, BppAttack exhibits merely 12.5% ASR with label flipping and training control without optimization by the proposed components. As shown in Table 3g, MultiBpp attains an ASR of 76.6% when quantized with the intensity of $24 : 48 : 8$ in RGB color channels. Therefore, the BppAttack can be optimized as effective clean-label poison-only attacks with higher ASR. According to Table 3{e, i, j}, the red and green channels demonstrate superior attack performance with 84.1% ASR by poisoning at the red channel and 72.2% ASR by poisoning at the green channel. The differential learning sensitivities imply that the model can infer that the feature in the red and green channels are more valuable. Tables 3{e,f} and {g,h} indicate that increasing the quantization step improves ASR. However, MultiBpp with the quantization intensity of $(36 : 72 : 12)$ yields a lower ASR of 57.7%, compared to 60.0% achieved with $255 : 255 : 12$. We hypothesize that the learning effectiveness of the poisoning feature is not solely influenced by the quantization step. Specifically, in scenarios like {f,h}, the model needs to focus on features from all three channels when learning under the configuration of h, whereas it only needs to attend to feature from one channel when learning the trigger feature. Experiments of component C upon {Badnets-C, Blended-C} can be seen in **Appendix F**.

**Collaborative effect of our components on ASR enhancement :** As depicted in Table 4, the positive ASRs of attacks occur when optimized by components A&C because optimal improvement in ASR cannot be achieved when consideration is given to stealthiness. Taking Badnets-C for example, the ASR decreases from 86.15% to 77.67% as Component B slightly reduces the effect of Component B on ASR enhancement. Compared to the ASR of vanilla, solely applying Component

Table 4: Performance of optimized attacks upon CIFAR-10 with 1% samples poisoned.

| Additive Components | Poisoning Rate $\alpha = 1\%$ | | | | Poisoning Rate $\alpha = 2.5\%$ | | | |
|---|---|---|---|---|---|---|---|---|
| | Badnets-C | | Blended-C | | Badnets-C | | Blended-C | |
| Method | ASR | BA | ASR | BA | ASR | BA | ASR | BA |
| Vanilla | 20.47 | **94.50** | 53.41 | **94.90** | 34.09 | **94.88** | 52.03 | **94.22** |
| + Component A | 70.03 | 94.16 | 70.65 | 93.93 | 73.19 | 93.19 | 85.15 | 93.70 |
| + Component B | 21.33 | 94.17 | 57.89 | 94.13 | 70.38 | 93.43 | 74.12 | 94.07 |
| + Component C | 38.67 | 94.47 | 60.46 | 94.16 | 45.59 | 94.31 | 74.77 | 93.92 |
| + Components A&B | 67.47 | 93.71 | 75.00 | 93.71 | 73.59 | 94.18 | 81.63 | 93.41 |
| + Components A&C | **86.15** | 93.90 | **84.13** | 93.97 | **89.85** | 93.65 | **94.32** | 94.11 |
| + Components B&C | 58.57 | 94.03 | 70.66 | 94.03 | 70.75 | 93.68 | 85.20 | 93.78 |
| + Components A&B&C | 77.67 | 94.01 | 77.51 | 94.11 | 84.49 | 93.45 | 87.54 | 93.76 |

B will not cause the reduction of ASR (from 20.47% to 21.33%). Furthermore, when applied at a higher poisoning rate, Component B can improve nearly 20% ASR of Badnets-C (Blended-C) from 34.09% to 52.03% (from 52.03% to 74.12%), respectively. The underlying cause may reside in the diminished competition between triggers and the target-class features, which is induced by the high similarity between triggers and benign images.

Furthermore, all components can be strategically integrated into diverse PCBAs, enabling tailored solutions that balance ASR and stealthiness enhancement for specific attack requirements. For invisible attacks such as Narcissus (Zeng et al. [2023a]), applying components A&C is enough. For example, we achieve a **new SOTA performance** in Backdoor Attack based on the SOTA attack (Narcissus). By poisoning merely 2 images (poison rate = 0.00004), the optimized Narcissus achieves 96.12% ASR and 95.10% BA in CIFAR-10 with 0 as the target-label. Guides of applicability to recent PCBAs can be seen in **Appendix J**.

Table 5: Performance of our methods on ASR when defended by defense methods.

| Attack | Method | Original | ABL | AC | FP | I-BAU | NC | RNP | FST |
|---|---|---|---|---|---|---|---|---|---|
| | random | 18.8 | 0 | 18 | 14.4 | 8.0 | 18.8 | 10.5 | 22 |
| | forget | 52.9 | 8.4 | 36.3 | 31 | 17.3 | 52.9 | 8.8 | 54.7 |
| | + Component A | 56.2 | **14** | 47.7 | 36.5 | 27.9 | 56.2 | 36.3 | 62.4 |
| Badnets-C | + Component B | 37.9 | 3.5 | 25.6 | 20.2 | **32.5** | 37.9 | 34.9 | 40.5 |
| | + Component C | 53.7 | 5.5 | 28.4 | 28.9 | 7.4 | 1.0 | 24.1 | 50.4 |
| | + Components B&C | 54 | 0.2 | 51.8 | 25.9 | 5.9 | 54 | 0 | 37.2 |
| | + Components A&C | **87.5** | 1.6 | **68** | 51.2 | 14.8 | 81.2 | 47.6 | 86.4 |
| | random | 57.6 | **15.1** | 52.4 | 39.8 | 28.3 | 57.6 | 27.1 | 42.8 |
| | forget | 76.1 | 9.2 | 73.2 | 63.5 | 7 | 76.1 | 0 | 64.1 |
| | + Component A | 77.9 | 3.9 | 76.9 | 64.3 | 20.0 | 77.9 | 36.1 | 69.5 |
| Blended-C | + Component B | 71.7 | 3.4 | 67.7 | 60.1 | 14.3 | 71.7 | 19.7 | 58.6 |
| | + Component C | 74.8 | 6.1 | 62.9 | 74.5 | **48.2** | 74.8 | 0 | 64.2 |
| | + Components B&C | 91 | 8.9 | 85.5 | 92.8 | 22 | 91 | 71.7 | 84.3 |
| | + Components A&C | **97.1** | 1.8 | **93.9** | **98.5** | 46.1 | **96** | **97.3** | **90.9** |

**Effect of Our Components on Backdoor Defense** We consider {ABL: Anti-backdoor Learning (Li et al. [2021b]), AC: Activation Clustering (Chen et al. [2018]), FP: Fine-pruning (Liu et al. [2018]), I-BAU: Implicit Hypergradient (Zeng et al. [2022]), NC: Neural Cleanse (Wang et al. [2019]), RNP: Reconstructive Neuron Pruning (Li et al. [2023c]), FST: Feature Shift Tuning (Min et al. [2023])} to analyze the impact of our methods on existing defense methods. All results are evaluated on CIFAR-10 by poisoning 3% samples at the clean-label setting. According to Table 5, each component exhibits a positive influence upon Badnets-C and Blended-C when defended by various defense methods in most cases for ASR. Specifically, Badnets-C enhanced by components A&C achieve 87.5% ASR, which is 68.7% higher than the original Badnets-C. Furthermore, the effectiveness of backdoor defenses sometimes depends mainly on the characteristics of the backdoor attacks and defense methods themselves. Badnets-C and Blended-C fail to penetrate the ABL. In such a case, the attacks optimized by our method also remain futile with ASR less than 1.8%. **In general, our methods outperform or keep the performance of the original attacks upon ASR when defended by defense methods**. Supplementary experiments are provided in **Appendix H**.

## 3.2 Ablation Study

**Effect of trigger scales:** BlendXs ($X \in \{20, 24, 28, 32\}$) are denoted to explore the effect of trigger scale X on the optimal combination in Component A for ASR enhancement. According to Table

Table 6: Performance of BlendXs upon CIFAR-10 with 1% samples poisoned.

| Sample Selection | | | Blend32 | | Blend28 | | Blend24 | | Blend20 | |
|---|---|---|---|---|---|---|---|---|---|---|
| Type | no. | Selection | ASR | BA | ASR | BA | ASR | BA | ASR | BA |
| Bench | a | Random | 53.44 | 94.95 | 48.47 | 94.69 | 39.45 | **94.89** | 18.15 | 94.68 |
| | b | Loss | 60.93 | **94.89** | 58.85 | 94.77 | 54.11 | 94.63 | 39.78 | 94.73 |
| | c | Gradient | 60.32 | 94.24 | 58.82 | **94.87** | 54.21 | 94.63 | 36.09 | **94.96** |
| | d | Forget | **70.53** | 94.59 | **79.49** | 94.33 | **72.01** | 94.70 | **66.80** | 94.15 |
| Ours | e | Res-$log$ | 82.34 | **94.73** | 82.69 | 94.56 | 76.42 | **94.95** | 67.49 | 94.48 |
| | f | Res-$x$ | 82.31 | 94.31 | 80.85 | **94.66** | **76.49** | 94.36 | **70.65** | **94.76** |
| | g | Res-$x^2$ | **84.88** | 94.38 | **83.85** | 94.55 | 75.42 | 94.60 | 68.00 | 94.66 |

6, the optimal selection strategies for {Blend32, Blend28, Blend24, Blend20} are {Res-$x^2$, Res-$x^2$, Res-$x$, Res-$x$}. As discussed in **Section 2.1**, Res-$x^2$ puts more emphasis on category diversity than Res-$x$. The model poisoned by Blend20 can learn the backdoor feature by focusing on a smaller area compared to Blend32, thereby decreasing the interference from the feature in other classes. Therefore, the significance of Forgetting Event is enhanced. According to the increasing gaps in ASR from Blend20 to Blend32, **larger trigger scales highlight the significance of Category Diversity.**

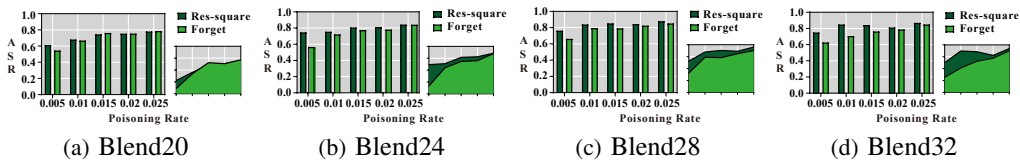

(a) Blend20     (b) Blend24     (c) Blend28     (d) Blend32

Figure 5: BlendX optimized by Component A (Res-$x^2$) with different poisoning rates.

**Effect of the poisoning rate :** According to Figure 4, the ASR gaps between the two methods upon {Blend20, Blend24, . . . , Blend32} gradually increase when the poisoning rate decreases from $0.25\%$ to $0.05\%$. Therefore, the superiority of our method becomes more pronounced when poisoning fewer samples, in which situation the significance of sample selection is highlighted. Furthermore, **Component A consistently outperforms Forgetting Event across various poisoning rates.**

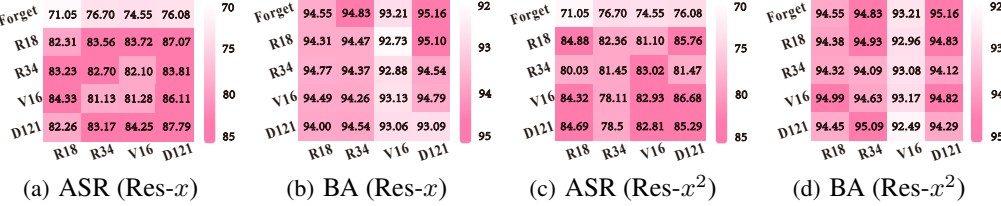

(a) ASR (Res-$x$)     (b) BA (Res-$x$)     (c) ASR (Res-$x^2$)     (d) BA (Res-$x^2$)

Figure 6: Performance of Component A upon Blended-C with different model structures. Row: models for pretraining. Column: victim models. 'Forget' indicates the results of poisoning samples selected based on the Forgetting Event calculated with Resnet18 (R18) as the model for pretraining.

**Effect of Model Structure :** We use {R18, R34, V16, D121} to represent {Resnet18, Resnet34, VGG16, DenseNet121}. As shown in Figure 5, Component A consistently outperforms Forgetting Event regardless of the concrete structure used in different stages. Most models optimized by Component A can get $10\%$ ASR improvement over the current SOTA metric (Forgetting Event). Furthermore, BAs of Blended-C remain stable around $94\%$. Therefore, **Component A can transfer across different model structures in both the pretraining stage and the training stage.**

## 4   Conclusion & Limitation

Current attacks usually handle the sample selection and triggers in isolation, leading to severely limited improvements on both ASR and stealthiness. Consequently, it is challenging to exhibit satisfactory performance when simply converted to PCBAs. A set of generalized components is proposed to improve both stealthiness and ASR of attacks to achieve effective PCBAs by sufficiently exploring the bi-directional collaborative relations between the sample selection and triggers, which can retain generalization ability upon various attacks. At the end, we list the limitations in the paper as follows: 1) The approach of integrating components A and B is rudimentary, and we will explore more scientific methods in future research. 2) Further research on exploring the collaborative relations between the sample selection and triggers remains necessary.

# 5 Acknowledgments

This work was supported in part by the NSFC fund (NO. 62206073, 62176077), in part by the Shenzhen Key Technical Project (NO. JSGG20220831092805009, JSGG20220831105603006, JSGG20201103153802006, KJZD20230923115117033, KJZD20240903100712017), in part by the Guangdong International Science and Technology Cooperation Project (NO. 2023A0505050108), in part by the Shenzhen Fundamental Research Fund (NO. JCYJ20210324132210025), and in part by the Guangdong Provincial Key Laboratory of Novel Security Intelligence Technologies (NO. 2022B1212010005), and in part by the Natural Science Foundation of Shenzhen General Project under Grant JCYJ20240813110007010, in part by the Natural Science Foundation of Guangdong Province under Grant 2023A1515010893, in part by the Shenzhen Pengcheng Peacock Startup Fund.

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

# A Related Work

In backdoor attacks, the adversary aims to embed a designed trigger in the victim model. Therefore, the poisoned models misclassify the trigger-embedded samples to the predefined target label (Gu et al. [2017], Chen et al. [2017]) while maintaining high accuracy for unaltered inputs. Multiple backdoor attacks prove their effectiveness in multimodal learning (Wang et al. [2024], Han et al. [2024]), federated learning (Li et al. [2023b], Chen et al. [2023]), diffusion model (Chou et al. [2023], Li et al. [2024a]), dataset distillation (Liu et al. [2023]), and other scenarios (Zhao et al. [2024]).

Among current backdoor attacks, Poison-only Backdoor Attacks (PBAs) have attracted huge attention given their widespread use and ease of construction in real-world scenarios (Li et al. [2021a],Qi et al. [2023]). PBAs poison the models by merely manipulating the training dataset, in which the effectiveness of attacks hinges on Trigger Design and Sample Selection.

## A.1 Trigger Design

Simply designed visible triggers in traditional attacks (Gu et al. [2017], Chen et al. [2017] can be effectively detected by humans and machines. Therefore, the adversary relies on the design of invisible triggers and physical triggers to ensure the stealthiness of the attacks.

In computer vision (CV), invisible triggers involve incorporating minor perturbations by tweaking the pixel values and positions of the original image (Bai et al. [2022]). The constraint of invisibility poses a significant limitation to achieving high ASR in the clean-label poison-only setting. Wenger et al. [2022] introduces natural triggers based on the hypothesis that there may be naturally occurring physically colocated objects already present in popular datasets such as ImageNet. Furthermore, some attacks(Lin et al. [2020], Zeng et al. [2023a]) propose triggers formulated from a combination of existing benign features to bypass the backdoor defense methods.

Efforts to overcome the dilemma frequently result in unsatisfactory performance (e.g., high poisoning rates, ineffective backdoor embeddings, limited transferability, and weakened robustness). For instance, Wang et al. [2022] introduces BppAttack, a stealthy attack that leverages image quantization and dithering to induce triggers into victim models. Given the constrained effectiveness of imperceptible modifications, adversaries struggle to enhance the ASR by employing adversarial training combined with label flipping. Recently, (Gao et al. [2024]) formulates a bi-level optimization problem to balance the conflict of ASR and stealthiness with sparsity and invisibility constraints. The upper-level optimization problem aims to minimize the loss on poisoned samples by optimizing the trigger. Meanwhile, the lower-level problem focuses on minimizing the loss across all training samples through the optimization of model weights, which deviates from a poison-only attack.

**Summary** Current PBAs primarily focus on the design of triggers, leading to multiple triggers that exhibit unique advantages under different metrics (e.g., design complexity, feature intensity, the ability to bypass defenses, stealthiness, and dataset dependency). Therefore, **it is valuable to explore generalization optimization strategies to enhance various triggers on both ASR and stealthiness.** Additionally, **current research overlooks the effect of sample selection in the design process.**

Although state-of-the-art backdoor attacks (Zeng et al. [2023a], Gao et al. [2024] currently manage to design potent invisible triggers through steps like training trigger generators, they come at the cost of substantial training overhead and the requirement for comprehensive knowledge of the entire dataset. **Exploring the enhancement of the traditional attacks via simple yet effective approaches represents a research topic worthy of in-depth investigation.**

## A.2 Sample Selection

Clean-label backdoor attacks are seen as the stealthiest attacks, as adversaries can only poison samples from the target class without changing their labels. The dilemma of unsatisfactory ASR of current PBAs that merely depend on the trigger design led to the research study of sample selection. Gao et al. [2023] reveals differential sample importance and selects "hard" samples via three metrics (e.g., Forgetting Event (as depicted in **Section 2.1**), Loss Value, and Gradient Norm) to enhance the PBAs. The poisoned models tend to learn the implicit projection between the trigger feature and the target label to evade the difficulty of the original classification upon such "hard" samples. Details of Loss Value and Gradient Norm can be seen as follows.

**Loss Value**  Given a benign model $f_\theta$ (trained on the benign training set $D_{tr}$), the loss value of model on sample $(x_i, y_i)$ can be represented as $L(f_\theta(x_i), y_i)$. We choose samples with the greatest $\alpha * |D_{tr}|$ values in the subset $D_t$ are chosen for poisoning:

$$D_s = arg \max_{D_s \subset D_t} \sum_{(x_i, y_i) \in D_s} L(f_\theta(x_i), y_i). \tag{6}$$

**Gradient Norm**  Given a benign model $f_\theta$ (trained on the benign training set $D_{tr}$), the $l_2-$ gradient norm of model on sample $(x_i, y_i)$ can be represented as $||\nabla_\theta L(f_\theta(x_i), y_i)||_2$. We choose samples with the greatest $\alpha * |D_{tr}|$ values in the subset $D_t$ are chosen for poisoning:

$$D_s = arg \max_{D_s \subset D_t} \sum_{(x_i, y_i) \in D_s} ||\nabla_\theta L(f_\theta(x_i), y_i)||_2. \tag{7}$$

Han et al. [2024] further improves the efficiency of attacks based on an optimized backdoor gradient-based score. Moreover, Hayase and Oh [2022] formulates sample selection as a bi-level optimization problem: construct strong poison examples that maximize the ASR. Furthermore, some scientists propose novel sample selection methods based on poisoning masks (Zhu et al. [2023]), confidence-based scoring (Wu et al. [2023]), and high-frequency energy (Xun et al. [2024]).

**Summary**  Current research on sample selection focuses on designing new metrics or training derivations to construct data-efficiency attacks, **overlooking the synergistic effect between triggers and sample selection on ASR enhancement.** Meanwhile, **current methods overlook the effect of sample selection on stealthiness enhancement.**

## B  Preliminaries

### B.1  Model Training

The model output function of the image classification can be denoted by $f_\theta : X \to Y$, where $x \in X = \{0, 1, \ldots, 255\}^{C \times H \times W}$ represents an image domain, $Y = \{y_1, y_2, \ldots, y_k\}$ is a set of k classes, and $\theta$ denotes the parameters that a DNN learned form the begin training dataset $D_{tr} = \{(x_i, y_i)\}_{i=1}^N$. The benign training with $D_{tr}$ can be seen as a single-level optimization problem. The optimization seeks a model $f_\theta$ by solving the following problem during training:

$$\min_\theta L(D_{tr}, f_\theta) = \sum_{i=1}^{N_{tr}} l(x_i, y_i, f_\theta), \tag{8}$$

where $l$ is the loss function (e.g., the cross-entropy), and $(x_i, y_i) \in D_{tr}$.

### B.2  Poison-only Clean-label Backdoor Attacks

#### B.2.1  Attack Knowledge

In a poison-only backdoor attack, an adversary has access to the original training dataset $D_{tr}$ and is allowed to inject the pre-defined trigger into a small subset of the training set. Specifically, attacks can be called clean-label attacks if the adversary does not change the ground-truth label of the original data. Furthermore, the adversary has no knowledge and the ability to modify other training components (e.g., loss functions, model architecture, training schedule, optimization algorithm, etc). Consequently, attackers can only influence model weights through data poisoning. The latent connection between the trigger and the target label is learned only during the training process.

#### B.2.2  Attack Workflow

We detail the workflow of poison-only clean-label backdoor attacks to formalize the theoretical foundations. How to generate the poisoned dataset $D_p$ is the cornerstone of the attack. Details about the attack, knowledge of poison-only clean-label backdoor attacks can be seen at Appendix B. We remark on the important evaluation criteria at the following steps.

**Step 1: Select samples to be poisoned (by attackers).** $D_p$ consists of two disjoint parts. Given a target label $y_t$, a subset $D_s$ is selected from target-label set $D_t = \{(x_i, y_i)|(x_i, y_i) \in D_{tr}, y_i = y_t\}$ to be poisoned and the remain benign samples can be denoted as $D_b = D_{tr} \backslash D_s$. Here we define a binary vector $M = [M_1, M_2, \ldots, M_{|D_{tr}|}] \in \{0, 1\}^{|D|}$ to represent the poisoning selection. Specifically, $M_i = 1$ indicates that $x_i$ is selected to be poisoned while $M_i = 0$ means the benign sample. We denote $\alpha := \frac{|D_s|}{|D_{tr}|}$ as the poisoning rate. Note that most existing backdoor attack methods randomly select $\alpha \cdot |D_{tr}|$ samples to be poisoned. $\alpha$ serves as a crucial indicator of stealthiness in poison-only attacks. Backdoor attacks are supposed to maintain a high attack success rate with $\alpha$ as small as possible to evade both machine and manual inspections.

**Step 2: Trigger Insertion (by attackers).** In computer vision applications, the adversary designs a trigger pattern $w$ by tweaking the pixel values and positions of the benign image. The generator of poisoned images can be denoted as $f_g : X \to X$. For example, $f_g(x) = (1 - m) * x + m * w$, where the mask $m \in [0, 1]^{C \times H \times W}$ representing the poison area of the trigger $w$ and $*$ representing the element-wise product. Therefore, given the target label $y_t$ in a clean-label attack, the generated poisoned training dataset could be denoted as $D_p = \{(x_i, y_i)|_{if \ m_i=0}, \ or \ (f_g(x_i), y_t)|_{if \ m_i=1}\}_{i=1}^{|D_{tr}|}$. For stronger stealthiness, the trigger $w$ is expected to be sufficiently invisible, which means the distance $L_D(f_g(x_i), x_i)$ should be small.

**Step 3: Model Training (by users).** Once the poisoned dataset $D_p$ is generated, users will train the poisoned DNN via the period described in section 3.1.1. The stealthiness and utility of backdoor attacks demand imperceptible dataset modifications, requiring the poisoned model $\tilde{f}_\theta$ to maintain high accuracy on benign test data. Otherwise, users would not adopt the poisoned model and no backdoor could be implanted. The accuracy on clean test set $D_{clean}$ can be computed by:

$$CleanACC = \frac{1}{N_{clean}} \sum_{i=1}^{N_{clean}} ACC(\tilde{f}_\theta(x_i), y_i) \tag{9}$$

where $N_{clean}$ means the number of clean test set. $(x_i, y_i) \in D_{clean}$ and $y_i$ is the ground-ruth label. $ACC(y_{pre}, y)$ will be set to 1 if $y_{pre} = y$ and 0 otherwise.

**Step 4: Activate the backdoor using the trigger during the inference stage (by attackers).** The attackers expect to activate the injected backdoor using the trigger $w$ defined in step 2. Given the poisoned model $\tilde{f}_\theta$, the Attack Success Rate (ASR) of a backdoor attack can be computed by:

$$ASR = \frac{1}{N_{clean}} \sum_{i=1}^{N_{clean}} ACC(\tilde{f}_\theta(f_g(x_i)), y_t) \tag{10}$$

where $N_{clean}$ means the number of clean test set $D_{clean}$. $f_g(x_i)$ represents the poisoned image on image $x_i$ and $y_t$ is the target label. $\tilde{f}_\theta$ and $ACC(y_{pre}, y)$ are defined in Step 3.

## C  Algorithms with other negative functions

In this chapter, we present the pseudocode implementation related to Res-X in our experiments. Here, the 'X' in Res-X correlates with the weight assigned to category diversity. Analogous to algorithmic complexity, Res-X places greater emphasis on the contribution of category diversity to the metrics compared to Res-log. As demonstrated in the main text, the optimal weight ratio is associated with trigger characteristics (e.g., the size of the poisoned region).

Currently, achieving the optimal integration of category diversity and forgetting events remains largely reliant on empirical approaches. Moving forward, we will delve deeper into uncovering more underlying patterns and focus on developing algorithms for automated integration.

**Algorithm 4** Metric Calculation with Negative Function $N_F$ at $O(n)$

---

**Input :** Train Dataset $D_{tr}$, Target Label $y_t$, Misclassification Events $N_e((x_i, y_i), y_m)$
**Output :** Calculated Metric of Samples
**for** image $(x_i, y_t) \in D_{tr}$ **do**
  $Num[y_m], Sum = 0$
  **for** $y_m \in Y$ **do**
    $Num[y_m] = Num[y_m] + N_e((x_i, y_t), y_m)$
    $Sum = Sum + Num[y_m]$
  **end for**
**end for**
**for** $y_m \in Y$ **do**
  $Cls[y_m] = 1 - \frac{Num[y_m]}{Sum}$
**end for**
**for** image $(x_i, y_t) \in D_{tr}$ **do**
  $Metric[x_i] = 0$
  **for** $y_m \in Y$ **do**
    $Metric[x_i] = Metric[x_i] + Cls[y_m] * N_e((x_i, y_t), y_m)$
  **end for**
**end for**

---

**Algorithm 5** Metric Calculation with Negative Function $N_F$ at $O(n^2)$

---

**Input :** Train Dataset $D_{tr}$, Target Label $y_t$, Misclassification Events $N_e((x_i, y_i), y_m)$
**Output :** Calculated Metric of Samples
**for** image $(x_i, y_t) \in D_{tr}$ **do**
  $Num[y_m] = 0$
  **for** $y_m \in Y$ **do**
    $Num[y_m] = Num[y_m] + N_e((x_i, y_t), y_m)$
  **end for**
**end for**
**for** $y_m \in Y$ **do**
  $Sum = Sum + Num[y_m] * Num[y_m]$
**end for**
**for** $y_m \in Y$ **do**
  $Cls[y_m] = 1 - \frac{Num[y_m] * Num[y_m]}{Sum}$
**end for**
**for** image $(x_i, y_t) \in D_{tr}$ **do**
  $Metric[x_i] = 0$
  **for** $y_m \in Y$ **do**
    $Metric[x_i] = Metric[x_i] + Cls[y_m] * N_e((x_i, y_t), y_m)$
  **end for**
**end for**

---

**Algorithm 6** Metric Calculation with Negative Function $N_F$ at $O(e^n)$

---

**Input :** Train Dataset $D_{tr}$, Target Label $y_t$, Misclassification Events $N_e((x_i, y_i), y_m)$
**Output :** Calculated Metric of Samples
**for** image $(x_i, y_t) \in D_{tr}$ **do**
  $Num[y_m] = 0$
  **for** $y_m \in Y$ **do**
    $Num[y_m] = Num[y_m] + N_e((x_i, y_t), y_m)$
  **end for**
**end for**
**for** $y_m \in Y$ **do**
  $Sum = Sum + exp(-Num[y_m])$
**end for**
**for** $y_m \in Y$ **do**
  $Cls[y_m] = 1 - \frac{exp(-Num[y_m])}{Sum}$

**end for**
**for** image $(x_i, y_t) \in D_{tr}$ **do**
    $Metric[x_i] = 0$
    **for** $y_m \in Y$ **do**
        $Metric[x_i] = Metric[x_i] + Cls[y_m] * N_e((x_i, y_t), y_m)$
    **end for**
**end for**

## D  Gradient Magnitude Similarity Deviation

Images visually insensitive to triggers are selected by calculating the GMSD between benign images and poisoned images to conceal the trigger feature in the target-label feature. GMSD is a full-reference image quality assessment (FR-IQA) model that leverages pixel-wise gradient magnitude similarity (GMS) to quantify local image quality and the standard deviation of the global GMS map to quantify the final image quality. Specifically, the gradient magnitude is derived using the Prewitt filter, which estimates horizontal $x$ and vertical $y$ gradient components via convolution by the following kernels:

$$h_x = \begin{bmatrix} 1/3 & 0 & -1/3 \\ 1/3 & 0 & -1/3 \\ 1/3 & 0 & -1/3 \end{bmatrix}, \quad h_y = \begin{bmatrix} 1/3 & 1/3 & 1/3 \\ 0 & 0 & 0 \\ -1/3 & -1/3 & -1/3 \end{bmatrix} \tag{11}$$

Convolving $h_x$ and $h_y$ with the reference and distorted images yields the horizontal and vertical gradient images of $r$ and $d$. $m_r(i)$ and $m_d(i)$ represent the gradient magnitudes of $r$ and $d$ at location $i$, which can be computed as follows:

$$m_r(i) = \sqrt{(r \otimes h_x)^2(i) + r \otimes h_y)^2(i)}, \quad m_d(i) = \sqrt{(d \otimes h_x)^2(i) + d \otimes h_y)^2(i)} \tag{12}$$

where symbol "$\otimes$" denotes the convolution operation. The gradient magnitude similarity (GMS) map is computed based on the gradient magnitude images $m_r(i)$ and $m_d(i)$ as follows:

$$GMS(i) = \frac{2m_r(i)m_d(i) + c}{m_r^2(i) + m_d^2(i) + c} \tag{13}$$

where $c$ is a positive constant that supplies numerical stability. Gradient Magnitude Similarity Mean (GMSM) serves as the local quality map (LQM) of the distorted image $d$ with average pooling applied to assume that each pixel has the same importance in estimating the overall image quality:

$$GMSM = \frac{1}{N} \sum_{i=1}^{N} GMS(i) \tag{14}$$

where $N$ is the total number of pixels in the image. Clearly, a higher GMSM score means higher image quality. Based on the idea that the global variation of image local quality degradation can reflect its overall quality, Gradient Magnitude Similarity Deviation (GMSD) is proposed to compute the standard deviation of the GMS map as the final IQA index:

$$GMSD = \sqrt{\frac{1}{N} \sum_{i=1}^{N} (GMS(i) - GMSM)^2} \tag{15}$$

GMSD serves as a quantitative measure of the spatial distribution of distortion severity within an image. Specifically, higher GMSD values indicate a wider range of distortion magnitudes across local regions, which correlates with degraded perceptual quality due to the exacerbated spatial inconsistency of degradation effects.

## E  Human Visual System

Computers encode image colors based on the three primary color channels (RGB). However, current design of triggers neglects the differences in human visual perception (Land and McCann [1971]) and machine representation. Therefore, knowledge of the human visual system (HVS) can assist adversary in more scientifically leveraging the disparities between the human eye and machine systems to enhance the stealthiness and functionality of triggers in backdoor attacks.

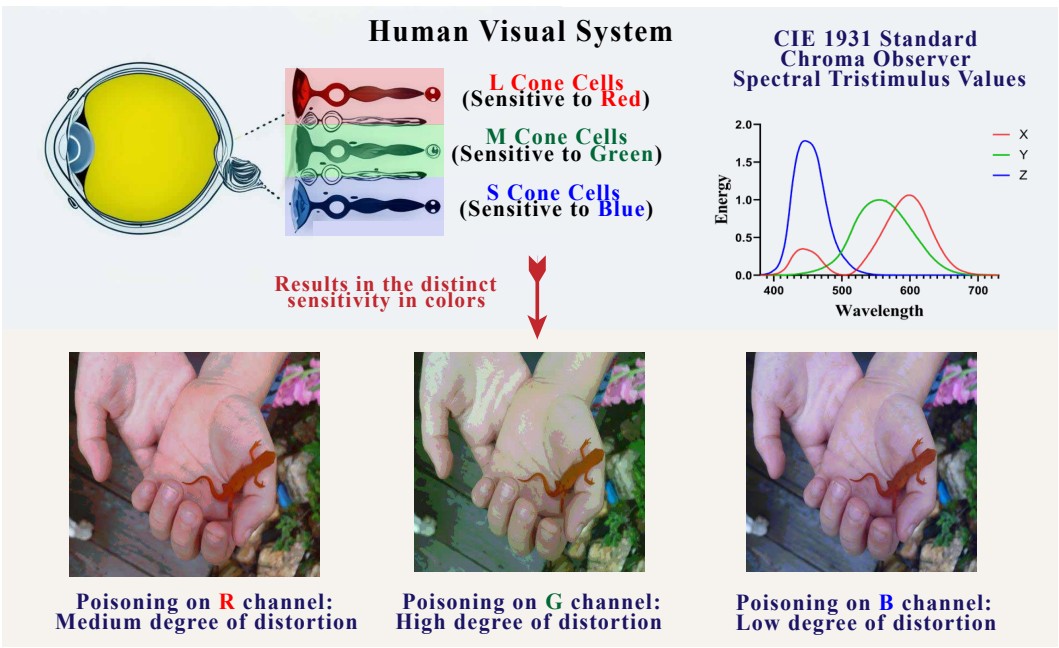

Figure 7: Distinct Sensitivity to Colors in Human Visual System.

## E.1 Distinct Sensitivity to RGB

The human retina contains three types of cone cells, each playing a crucial role in color vision by being sensitive to different wavelengths of light. These three types of cone cells work together to provide us with color vision. Each type of cone cell contains a different photopigment that is sensitive to a specific range of wavelengths. When light enters the eye and stimulates these cone cells, they send signals to the brain, which then processes this information to produce our perception of color.

**Long-Wavelength Sensitive (L) Cone Cells:**

- These cone cells are most responsive to long-wavelength light, with a peak sensitivity around 560 nm, which corresponds to the yellow-green region of the visible spectrum.
- They are often referred to as "red" cone cells because of their relative sensitivity to longer wavelengths, although their peak is not precisely at the red end of the spectrum.
- L cone cells are abundant in the retina and are essential for distinguishing between colors in the red-yellow-green range.

**Medium-Wavelength Sensitive (M) Cone Cells:**

- M cone cells have their peak sensitivity around 530 nm, in the green region of the spectrum.
- These cone cells are crucial for perceiving colors in the green range and are involved in color discrimination tasks that require distinguishing between different shades of green and yellow.
- Together with L cone cells, M cone cells form the basis for our perception of a wide range of colors in the visible spectrum.

**Short-Wavelength Sensitive (S) Cone Cells:**

- S cone cells are most responsive to short-wavelength light, with a peak sensitivity around 420 nm, which corresponds to the blue-violet region of the spectrum.
- They are often referred to as "blue" cone cells and are essential for perceiving colors in the blue range.
- S cone cells are less abundant in the retina compared to L and M cone cells, but they play a critical role in our ability to distinguish between colors that have a blue component.

The RGB color system is based on the three primary colors of human vision. Experiments have revealed that when certain spectral colors are represented using the color-matching functions of the RGB color system, negative values emerge. This implies that there are spectral colors that cannot be expressed using the visual primary colors RGB. Therefore, the XYZ color space system in the International Commission on illumination (CIE-XYZ) is introduced to address the dilemma.

We use $\{R, G, B\}$ to represent value of pixels in the three color channels $\{x^R, x^G, x^B\}$. The core objective of the CIE-RGB system is to establish an anchored relationship between color and physical parameters, ensuring a one-to-one correspondence between color perception and tristimulus values. Its design focuses on color appearance through the proportioning of the three primary colors, rather than directly quantifying the sensitivity of the human visual system. The phenomenon that human eyes are most sensitive to green light (555nm) is reflected in the subsequent CIE-XYZ system through the luminance function $f_Y = 0.2126R + 0.7152G + 0.0722B$, but this weight distribution is a characteristic of the CIE-XYZ system, not the original design of the CIE-RGB system.

In 1931, CIE standardized conversion relationships between the two systems to resolve the RGB system's negative value issue, guaranteeing positive tristimulus values in XYZ. Converting RGB values to CIE-XYZ tristimulus values follows a standardized process and the overall process of selecting samples can be outlined step-by-step below:

**Step 1: Normalize CIE-RGB values.** Step 1 aims to convert the value of image $(R, G, B)$ to the range $[0, 1]$ :

$$x_{norm}^c = \frac{x^c}{R + G + B}, c \in \{R, G, B\} \tag{16}$$

Specifically, we use $\{r, g, b\}$ to represent the normalized result $\{x_{norm}^R, x_{norm}^G, x_{norm}^B\}$.

**Step 2: Convert normalized CIE-RGB to normalized CIE-XYZ.** The conversion formulas of chromaticity coordinate conversion can be denoted as:

$$\begin{cases} X = (0.490r + 0.310g + 0.200b) \, / \, (0.607r + 1.132g + 1.200b) \\ Y = (0.117r + 0.812g + 0.010b) \, / \, (0.607r + 1.132g + 1.200b) \\ Z = (0.000r + 0.010g + 0.990b) \, / \, (0.607r + 1.132g + 1.200b) \end{cases} \tag{17}$$

CIE 1931 Standard Chroma Observer Spectral tristimulus Values, abbreviated as CIE Standard Chroma Observer, characterizes human ocular spectral sensitivity across wavelengths, as depicted in Figure 7. Furthermore, humans exhibit limited sensitivity to blue light because the blue-sensitive cone cells comprise merely $5\%$ in the human visual system.

**Summary** Based on the above observations, it is appropriate to reassign the poisoning intensity of the trigger design with a particular enhanced poisoning intensity in the blue channel.

## F  Supplemental Experiments about Injection Intensities of Triggers

Table 7: Performance of Badnets attacks upon CIFAR-10 with 1% samples poisoned.

| Trigger | | | Poisoning Rate $\alpha = 1\%$ | | | | Poisoning Rate $\alpha = 2.5\%$ | | | |
|---|---|---|---|---|---|---|---|---|---|---|
| Pattern | | | Random | | Res-$x^2$ | | Random | | Res-$x^2$ | |
| Type | no. | Method | ASR | BA | ASR | BA | ASR | BA | ASR | BA |
| | a | 0:0:0 | **41.99** | 94.16 | **90.74** | 94.11 | **78.29** | 94.68 | **92.97** | **94.58** |
| | b | 1:1:1 | 12.13 | 94.23 | 70.00 | **94.13** | 34.09 | 94.88 | 74.63 | 94.36 |
| RGB | c | 2:2:2 | 10.42 | 94.08 | 60.79 | 93.81 | 37.37 | 94.48 | 80.04 | 94.24 |
| | d | 1:1:0 | 37.31 | **94.45** | 86.15 | 93.90 | 63.62 | 94.92 | 89.52 | 94.51 |
| | e | 2:2:0 | 20.50 | 94.31 | 83.08 | 94.10 | 71.80 | **94.97** | 90.36 | 94.29 |
| | f | 3:3:0 | **40.92** | 94.79 | **74.74** | 94.86 | **68.05** | 94.75 | **91.23** | 94.01 |
| B | g | 3:3:1 | 12.15 | 94.05 | 53.42 | **94.97** | 26.47 | 94.39 | 70.80 | 94.42 |
| | h | 3:3:2 | 28.80 | **94.96** | 60.85 | 94.63 | 49.75 | 94.62 | 68.49 | **94.52** |

As depicted in Table 7 {a,f}, Badnets attained a notably higher ASR when employing a black-and-white trigger compared to monochromatic triggers (all-black, all-white). Currently, the distinctive

nature of the black-and-white trigger poses a greater challenge in identifying appropriate images for trigger concealment with Component B. According to the results between b,c and d,e, incorporating more pronounced trigger features exclusively within the blue channel also increases ASR in Badnet attacks. Consequently, integrating robust features solely into the blue channel, which exhibits lower sensitivity to human perception, can solve the dilemma about the sample selection for stealthiness.

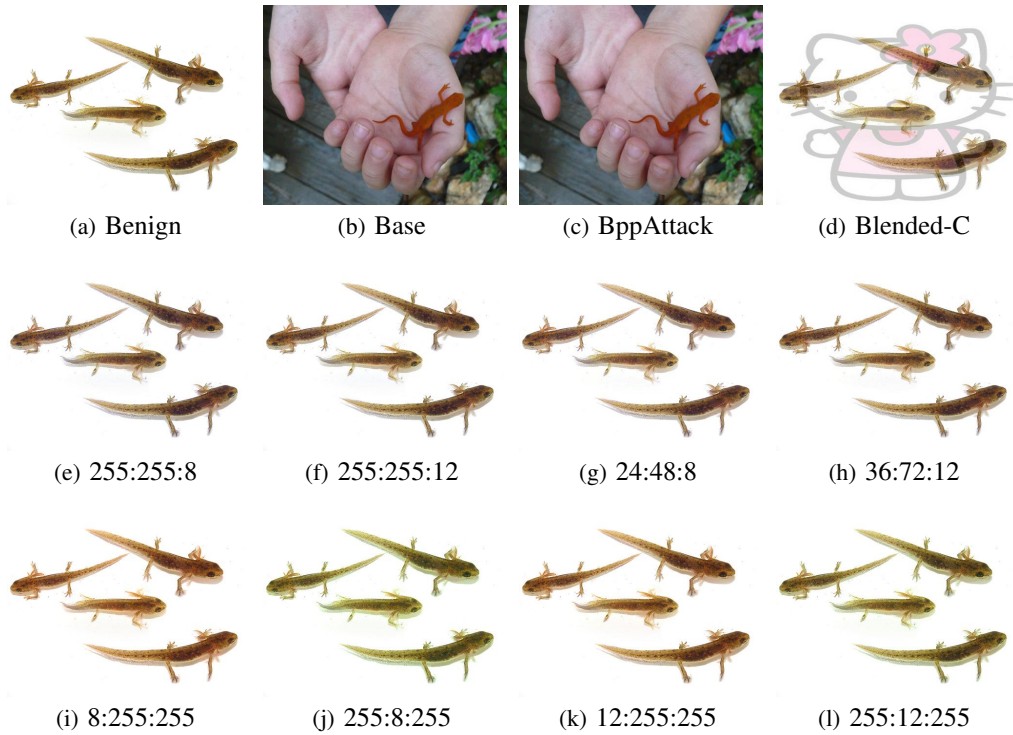

Figure 8: Visualizations of images in global-poisoning attacks. Compared to the benchmark (the first line), images that are visually insensitive to MultiBpp are selected in Component B. $N_R : N_G : N_B$ represent the distinct quantization intensity in $R : G : B$ channels.

As depicted in Figure 8, the original BppAttack randomly selects data for poisoning. To maintain the stealthiness of the trigger, BppAttack must adopt a smaller quantization step $(32 : 32 : 32)$, making it difficult to learn the trigger feature. We optimize the BppAttack based on two key observations. Firstly, current research on colorimetry reveals that the human visual system exhibits vastly different sensitivities to colors, as depicted in **Appendix E**. For example, we can observe that enhanced attacks by increasing the intensity of poisoning in the blue channel can still maintain invisibility to the human eye (Figure 8e) compared to enhanced attacks on other channels (Figure 8i, Figure 8j).

Different images exhibit different visual insensitivity to the specific trigger. For example, we can observe that the MultiBpp attack can still maintain more invisibility to the human visual system by poisoning images in Figure 8e compared to images in other images (e.g., image in Figure 8b). However, the image in Figure 8b is more visually insensitive for Blended-C compared to the images in Figure 8e. Therefore, the stealthiness of the trigger can be effectively preserved by carefully selecting appropriate samples based on the characteristics of the trigger pattern.

## G   Details of Experiment Setting

**Dataset and Model**   We conduct experiments on three benchmark datasets, including CIFAR-10, CIFAR-100, and Tiny-ImageNet. ResNet18 is the default model used to train the poisoned dataset. Among all datasets, the first class $(y = 0)$ is designated as the target class. The target class of each dataset is fixed across all the attacks adopting it. Standard augmentations are adopted on each dataset

Table 8: Hyperparameters and settings used in various datasets.

| Dataset | CIFAR-10 | CIFAR-100 | Tiny-ImageNet |
|---|---|---|---|
| # of Classes | 10 | 100 | 200 |
| Input Size | (3, 32, 32) | (3, 32, 32) | (3, 64, 64) |
| # of Images | 50000 | 50000 | 100000 |
| Target Class | 0 (Airplane) | 0 (Apple) | 0 (Goldfish) |
| Epochs | 200 | 200 | 400 |
| Optimizer | SGD (Stich et al. [2018]) | SGD (Stich et al. [2018]) | SGD (Stich et al. [2018]) |
| Augmentation | [Crop, H-Filp] | [Crop, Rotation] | [Crop, Rotation, H-Filp] |
| Model | Resnet18 | Resnet18 | Resnet18 |

to increase the model performance following existing training pipelines (He et al. [2016a], Tan and Le [2019]). Details of the dataset can be seen in Table 8.

**Attack Setup**  Three types of backdoor attacks {Badnets, Blended, BppAttack} are used as baselines to demonstrate the generalization ability of our components in {local high-intensity poisoning attacks, global medium-intensity poisoning attacks, global low-intensity poisoning attacks}.

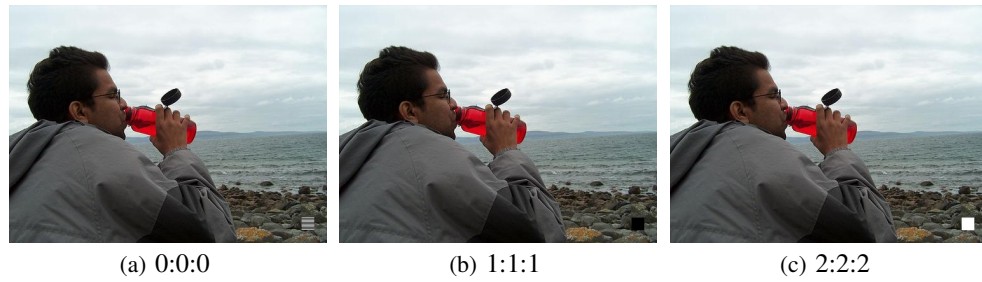

| (a) 0:0:0 | (b) 1:1:1 | (c) 2:2:2 |
|---|---|---|

Figure 9: Visualizations of different trigger patterns in Badnets attacks. Specifically, we use {0,1,2,3} to represent {black and white striped, all-black, all-white, vanilla} triggers. Futhermore, $N_R : N_G : N_B$ represent the distinct trigger pattern applied in $R : G : B$ channels.

As depicted in Figure 9, for BadNets attacks, a $3 \times 3$ random noise checkerboard pattern is utilized as the trigger in CIFAR-10 and CIFAR-100. For Tiny-ImageNet, a $9 \times 9$ is utilized as the trigger in BadNets attacks. {0, 1, 2, 3} represents the distinct {black and white striped, all-black, all-white, vanilla} trigger pattern and $N_R : N_G : N_B$ represent the distinct trigger pattern applied in $R : G : B$ channels, as depicted in Figure 8. The origin Badnets attack can be seen as attacks with whole-black triggers $(1 : 1 : 1)$. The Badnets trigger optimized by Component C can be represented as $(1 : 1 : 0)$. The experiments about Component A in Tables 1&2 follow the same setting as the original paper Gao et al. [2023], in which the Badnets trigger can be seen as $(0 : 0 : 0)$.

Secondly, for Blended attacks, a Hello-Kitty image is selected as the trigger and blended with the original images. $N_R : N_G : N_B$ represents the distinct trigger intensity applied in $R : G : B$ channels. The default of Blended attacks can be seen as attacks with a transparency parameter of $0.2 : 0.2 : 0.2$. The Blended trigger optimized by Component C can be represented as $(0.2 : 0.1 : 0.3)$.

Furthermore, in MultiBpp attacks, the ratio $(N_p^R : N_p^G : N_p^B)$ denotes the specific quantization configuration for poisoning intensity across the RGB channels. Notably, the default bit depth employed by BppAttack in the original study is set at 5, which, in the context of this paper, corresponds to a quantization ratio of $32 : 32 : 32$. Consequently, the "Base" scenario in our analysis refers to a quantization attack executed with the $32 : 32 : 32$ratio, excluding any training control mechanisms or label flipping operations inherent to the BppAttack methodology.

# H  Extended Ablation Study

Features in backdoor attacks can be classified into {trigger feature, target-label feature, feature in non-target classes}. Given the trigger feature is adjustable, the proposed components {Component A, Component B, Component C} mainly explore the potential of the inner relation between {trigger feature, feature in non-target classes}, {trigger feature, target-label feature} and {trigger feature, trigger feature}. Overall visualization can be seen in Figure 10.

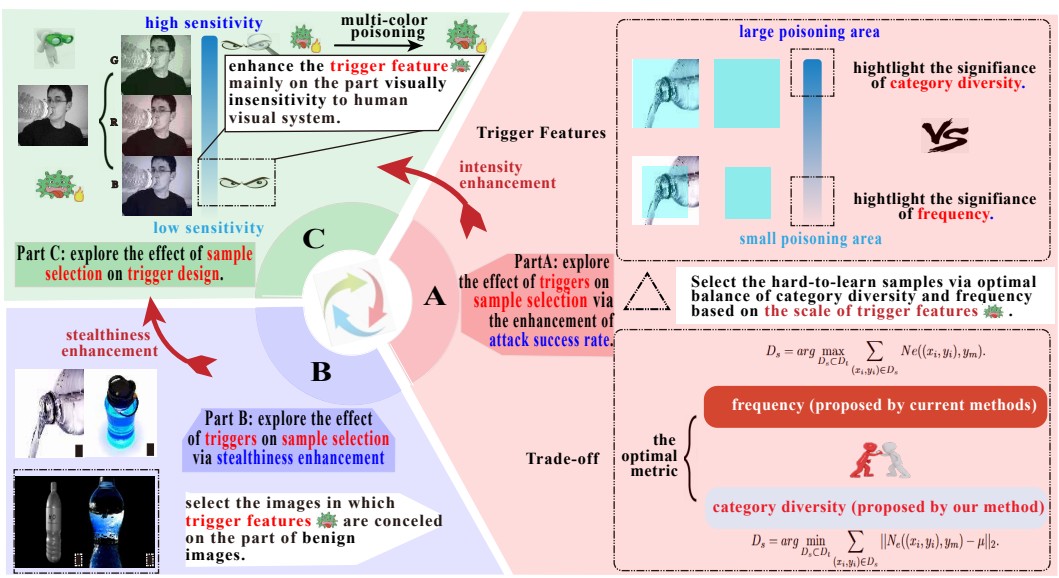

Figure 10: Overall visualization of our proposed components.

## H.1 Effect of Target Label

Table 9: Performance of Badnets-C with different target labels in CIFAR-100.

| *Target Label : 0* | | | *Target Label : 10* | | | *Target Label : 20* | | |
|---|---|---|---|---|---|---|---|---|
| Selection | ASR | BA | Selection | ASR | BA | Selection | ASR | BA |
| Forget | 59.39 | 78.21 | Forget | 85.4 | 78.11 | Forget | 59.4 | **78.8** |
| **Res-**$x$ | **80.48** | **78.25** | **Res-**$x$ | **91.68** | **78.12** | **Res-**$x$ | **73.48** | 78.5 |
| *Target Label : 30* | | | *Target Label : 40* | | | *Target Label : 50* | | |
| Selection | ASR | BA | Selection | ASR | BA | Selection | ASR | BA |
| Forget | 72.94 | 78.31 | Forget | 93.23 | **78.74** | Forget | 82.3 | **78.69** |
| **Res-**$x$ | **75.83** | **78.41** | **Res-**$x$ | **96.28** | 78.27 | **Res-**$x$ | **89.46** | 78.45 |
| *Target Label : 60* | | | *Target Label : 70* | | | *Target Label : 80* | | |
| Selection | ASR | BA | Selection | ASR | BA | Selection | ASR | BA |
| Forgetting Event | 38.78 | 78.5 | Forgetting Event | **81.96** | **78.56** | Forgetting Event | 88.46 | **78.61** |
| **Res-**$x$ | **46.57** | **78.68** | **Res-**$x$ | 79.51 | 78.55 | **Res-**$x$ | **89.1** | 78.32 |

**Result Analysis**   To explore the effectiveness of the proposed strategy (e.g., **Res-**$x$) on different target labels, we select labels ($y \in \{0, 10, 20, \dots, 80\}$) from CIFAR-100 and $20\%$ of the samples from the target class (representing $0.2\%$ of the total samples) are poisoned for Badnets-C.

As depicted in Table 9, Component A exhibits a higher ASR compared to the existing state-of-the-art metric, Forgetting Event (Forget), across an overwhelming majority of experimental conditions. Notably, a substantial variation in the efficacy of backdoor attacks and corresponding defensive filtering mechanisms is contingent upon the specific target class under consideration. To illustrate, the attack success rate of the Badnets model exhibits a stark contrast, registering at 46.57% when the target class is 60, yet surging to an impressive 96.28% when the target class is 40. Furthermore, the application of our method yields a notable enhancement of 21 percentage points in performance when the target class is 0, conversely experiencing a marginal decline of 3 percentage points when the target class is 70. **Therefore, Component A exhibits the widespread applicability and robust superiority upon ASR enhancement across diverse target labels.**

## H.2 Category Similarity

In CIFAR-10, the correspondence between y and the true labels is {0:airplane, 1:automobile, 2:bird, 3:cat, 4:deer, 5:dog, 6:frog, 7:horse, 8:ship, 9:truck}. Samples of class A but frequently misclassified as class B suggest a high level of similarity between A and B. As illustrated in Figure 11, significant variations exist in the similarity among different categories. For instance, the proportion of trucks

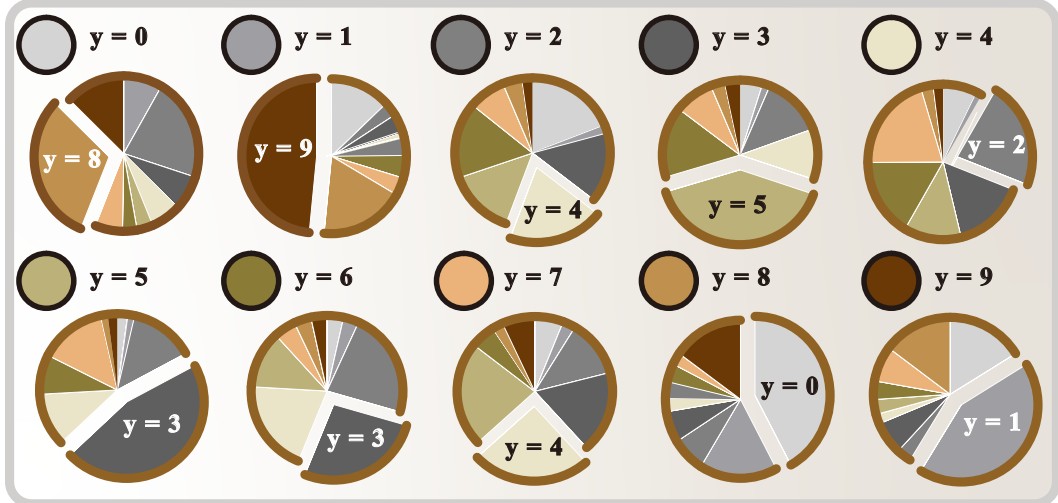

Figure 11: Category stability in CIFAR-10. We systematically arrange the proportions of misclassified categories across various data categories y, emphasizing the most prevalent category through white text highlighting. Above each visualization, the correct category corresponding to the pie chart, as well as its representative color in the context of other pie charts, is distinctly labeled.

(y=9) is substantially larger than that of birds (y=2). Therefore, automobiles (y=1) exhibit a much higher similarity to trucks than to birds in the pie chart representing automobiles (y=1). Furthermore, the similarity pattern displays symmetry. For the set y={0, 1, 2, 3, 4, 5, 8, 9}, the class with the highest proportion in its corresponding pie chart also dominates the pie chart of the corresponding class. Although y={6, 7} deviates from this trend, they still occupy the second-highest proportion in the corresponding pie charts y={3, 4}.

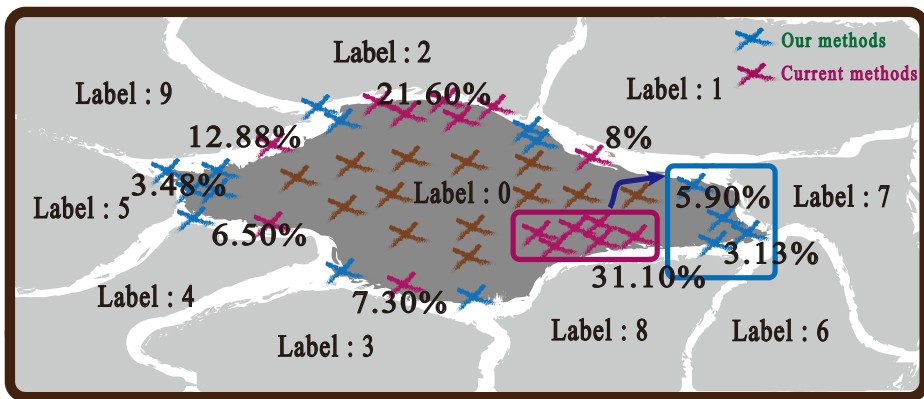

Figure 12: Visualization of misclassification results of label 0 in CIFAR-10. We use the edge lengths and associated numerical values to indicate the proportion of category information in Forgetting Event according to the pretraining stage.

Similar to entropy theory, samples belonging to class y (e.g., automobile) but frequently misclassified into a dissimilar class (e.g., bird) are more challenging to be learned by models and potentially more valuable than samples misclassified into a similar class (e.g., truck). For example, as depicted in Figure 12, samples belonging to {0: airplane} are often misclassified as {8: ship} (31.10%) rather than {8: frog} (3.13%) because of the distinct similarity. We hypothesize that samples misclassified as {8: frog} may be more informative and, therefore, should be considered in sample selection. Component B balances Forgetting Events and Category Diversity in sample selection. **Poisoning selected samples encourages the model to adopt shortcuts, facilitating the learning of the trigger feature.** {3, 5} ({cat, dog}) are both small-to-medium-sized animals, sharing high similarity in color and body shape, which collectively drives the model to prioritize learning of these two classes. In contrast, frogs exhibit lower similarity compared to other animal categories and do

not establish a synergistic relationship with them. Similarly, when juxtaposed with other modes of transportation (automobile, ship, truck), airplanes possess fewer common attributes with frogs, resulting in diminished model attention toward this category.

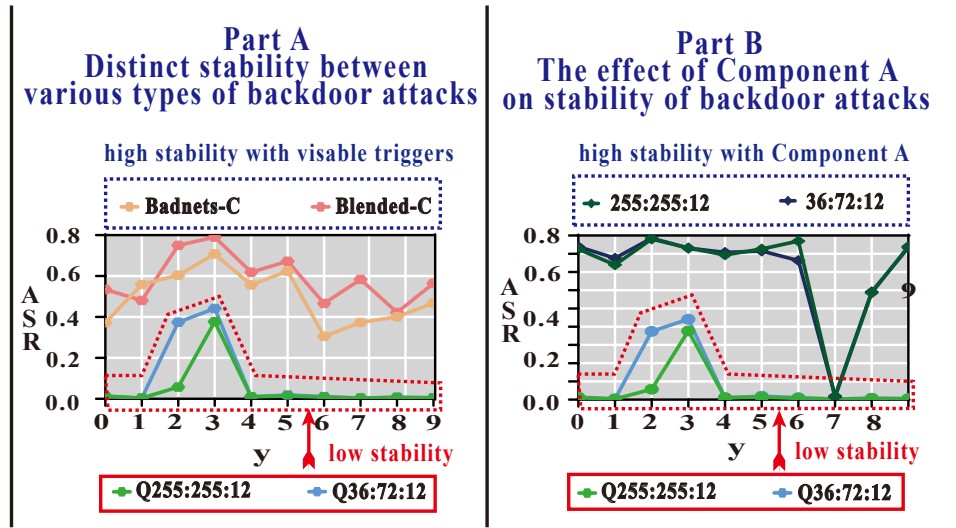

Figure 13: Inference of category similarity on backdoor attacks in CIFAR-10. Part A exhibits the effect of categories upon various backdoor attacks when randomly selecting samples to poison. Part B further explores the effect of Component A on the quantization attacks. Q-X denotes the simple quantization attack without Component A in which X is the quantization strength.

What is more, it is important to note that the similarity metrics depicted in Figure 11 represent the relative proportion of similarity to other classes rather than reflect the absolute magnitude of similarity between two classes. Consequently, the trigger feature embedded in classes with reduced model attention receives less focus, ultimately undermining the efficacy of backdoor attacks. As shown in Figure 13 Part A, Badnets-C and Blended-C exhibit superior attack performance (maxima) at y = {3, 5} (cat, dog). In contrast, they demonstrate inferior attack performance at y = {0, 6} (airplanes, frog). For quantization methods without Component A, the attack is entirely ineffective in scenarios other than y = {2, 3}. The efficacy of backdoor attacks exhibits substantial fluctuations tied to class characteristics and reflects inadequate stability when randomly selecting samples to poison.

In contrast, the merits of backdoor attacks with Component A manifest themselves in the following ways: (1) Component A can enhance the ASR of backdoor attacks. For example, with y = 0, the ASR of BadNet attacks with component A receives a 40 percentage point enhancement. (2) Component A can ensure the stability of backdoor attacks. The variability in attack success rates across different classes is notably mitigated when our methodology is implemented. In particular, for quantization attacks characterized by a weak trigger feature, component A leads the models to circumvent class-specific constraints and detect the embedded backdoor patterns. However, while an effective trigger strategy can attenuate, but not eliminate, the impact of class-related factors, quantization attacks remain ineffective in the context of y = 7 (horse).

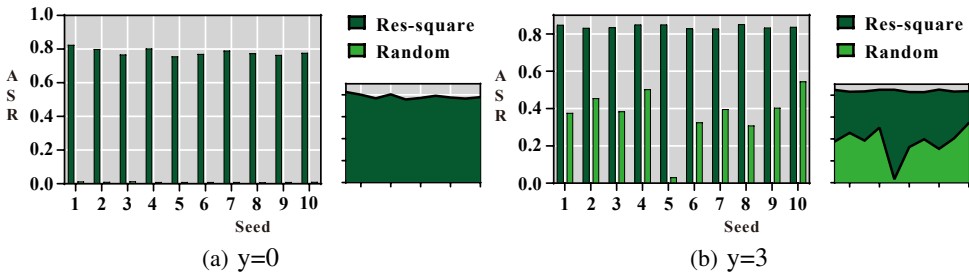

Figure 14: Stability of Badnets attacks by selecting samples using **Res-**$x$.

Figure 14 delineates the failure probabilities associated with backdoor attacks aimed at the aircraft class (y=0) and the cat class (y=3), alongside the attack efficacy following the implementation of our devised poisoning data selection methodology, referred to as **Res-$x^2$**. The term "seed" denotes the random seed employed to regulate and replicate stochastic processes. In experiments where poisoning data is chosen arbitrarily, the model consistently fails to acquire quantized backdoor patterns across all ten iterations within the attack configuration targeting the aircraft class. Conversely, in the scenario focusing on the cat class, the model achieves a 90% likelihood of acquiring a quantized backdoor feature, with a singular instance of failure observed under the condition of seed=5.

The aforementioned observations underscore that the model's capacity to assimilate backdoor features is contingent upon random variables, exhibiting a strong correlation with class-specific attributes. A judicious poisoning data selection strategy can markedly bolster the robustness of the attack. Upon integrating our proposed **Res-$x^2$** poisoning data selection strategy, the model demonstrates a flawless 100% success rate in learning backdoor features over twenty iterations, thereby significantly attenuating the detrimental influence exerted by the choice of target class on backdoor attacks.

### H.3 Applying our methods to poisoned-label backdoor attacks

In this section, we examine the applicability of these strategies to enhance poisoned-label backdoor attacks. In the poisoned-label scenario, the selection of poisoned samples is conducted across the entire training dataset rather than being confined to the target class. $\{0.05\%, 0.1\%\}$ of the total samples are poisoned for Badnets. We evaluate our plug-in methods (**Res-$x$**) against the standard version with random selection (dubbed 'Random').

Table 10: Performance of poison-label attacks in CIFAR-100 with different poisoning rates.

| *Poisoning Rate : 0.05%* | | | | *Poisoning Rate : 0.1%* | | | |
|---|---|---|---|---|---|---|---|
| Attack | Selection | ASR | BA | Attack | Selection | ASR | BA |
| | **Random** | **69.25** | **78.19** | | **Random** | **80.77** | **78.72** |
| Badnets | Forget | 7.00 | 78.26 | Badnets | Forget | **53.98** | 78.51 |
| | **Res-$x$** | **27.55** | **78.59** | | **Res-$x$** | **54.53** | **78.55** |
| | **Random** | 73.01 | **78.21** | | **Random** | 81.76 | 78.42 |
| Blend | Forget | **64.07** | 78.31 | Blend | Forget | **73.29** | **78.70** |
| | **Res-$x$** | 62.66 | **78.74** | | **Res-$x$** | 71.33 | 78.47 |

**Effect of Component A in dirty-label attacks :** As illustrated in Table 10, there is a {41.70% (69.25% - 27.55%), 10.35% (73.01% - 62.66%)} decrease compared to the Random upon ASR of the {Badnets, Blend} attacks when optimized by our method with 0.05% samples poisoned in CIFAR-10. Therefore, **Component A exhibits limited applicability within poison-label attack scenarios**.

However, remediation to the dirty label is unnecessary in our paper. We aim to optimize dirty-label attacks to clean-label attacks while preserving high ASR, rendering further optimization in dirty-label scenarios less critical in our paper. The reason will be explored in future work. We also provide an analysis of the phenomenon. Under clean-label settings with airplane as the target label, component A selects images of airplanes that least resemble airplanes based on the forgetting events with its category diversity. According to the phenomenon of models taking shortcuts discussed in another paper, models tend to rely on learning triggers as a shortcut to solve the hard task in the selected images. Therefore, component A gets higher ASRs because generic airplane features exert minimal interference from backdoor features. In contrast, under dirty-label settings, it is more effective to directly use cat images for poisoning instead of selecting airplane images that least resemble airplanes. The model faces the greatest difficulty in classifying cats as airplanes and resorts to backdoor shortcuts, resulting in higher ASRs.

### H.4 The effect of Component A in Tiny-Imagenet

**Analysis on Tiny-imagenet**  We conduct experiments on Tiny-Imagenet, which is a simplified version of Large Scale Visual Recognition Challenge 2016 Russakovsky et al. [2015], with ResNet-18 (He et al. [2016b]). We compare our plug-in methods against the standard version with random selection (dubbed 'vanilla') and existing sample selection strategies based on current metrics (such as

Table 11: Current methods on Tiny-ImageNet.

| Method | Metric | Badnets-C | Blended-C |
|---|---|---|---|
| Vanilla | BA | 57.50% | 57.27% |
| | ASR | 17.06% | 27.71% |
| Loss Value | BA | 57.17% | 57.49% |
| | ASR | 32.22% | 37.63% |
| Gradient Norm | BA | 57.69% | 57.82% |
| | ASR | 31.74% | 38.74% |
| Forgetting Event | BA | 57.60% | 57.48% |
| | ASR | **32.29**% | **40.59**% |

Table 12: Our methods on Tiny-ImageNet.

| Method | Metric | Badnets-C | Blended-C |
|---|---|---|---|
| res-log | BA | 57.03& | 57.24% |
| | ASR | 34.46% | 42.02% |
| res-linear | BA | 57.17% | 57.16% |
| | ASR | 32.22% | 41.48% |
| res-square | BA | 58.01% | 57.02% |
| | ASR | **38.96**% | **43.93**% |
| res-exp | BA | 57.60% | 57.58% |
| | ASR | 32.29% | 38.31% |

forgetting events, gradient norm, and loss value). Across all these attacks upon Tiny-Imagenet, $50\%$ of the samples from the target class (representing $0.25\%$ of the total samples) are poisoned, with the first class designated as the target class. Results can be seen in Tables 11&12.

Badnets-C with Res-square achieves $38.96\%$ ASR, which is $6.67\%$ higher than the current optimal metric (Forgetting Event). Blended-C with Res-square achieves $43.93\%$ ASR, which is $3.34\%$ higher than Forgetting Event. Furthermore, Badnets-C reaches optimal ASR when adopting the more aggressive Res-square strategy on Tiny-ImageNet instead of the optimal strategy (Res-log) when trained on CIFAR10. This indicates that in the Tiny-Imagenet dataset with more categories (200), Category Diversity should be highlighted when searching for the appropriate combination of Forgetting Event and Category Diversity in Component A. Most clean-label attacks exhibit ineffective performance in large datasets. For Tiny-ImageNet with 200 classes, the clean-label poisoning rate is constrained to be less than 0.005. In that case, **each part of the poisoning process must be meticulously designed, thereby highlighting the value of our proposed methods.**

### H.5 Supplemental Experiments on Backdoor Detection

In this section, we investigate the effectiveness of our components against existing clean-label backdoor attacks. Specifically, we select SIG (Barni et al. [2019a]) and CTRL (Li et al. [2023a]) as representative attacks for experimentation. SIG represents standard clean-label attacks, while CTRL exemplifies self-supervised learning (SSL) backdoor attacks in the clean-label setting. Other experimental configurations remain consistent with the experiments in **Section 3.2**.

Table 13: Performance of our methods on PCBAs when defended by defense methods.

| Defense Methods | SIG | | | | CTRL | | | |
|---|---|---|---|---|---|---|---|---|
| | original | | our method | | original | | our method | |
| | bASR | ASR | bASR | ASR | bASR | ASR | bASR | ASR |
| Anti-Backdoor Learning | 94.2 | 0.4 | **97.2** | 0 | 91.3 | 66.5 | **96.5** | **85.8** |
| Activation Clustering | 94.2 | 93.5 | **97.2** | **97.5** | 91.3 | 84.2 | **96.5** | **91** |
| Fine Pruning | 94.2 | 61.6 | **98** | **88.4** | 91.3 | 94.9 | **97.2** | **99.2** |
| Adversarial Unlearning | 94.2 | 8.9 | **97.2** | **42.4** | 91.3 | 39 | **96.5** | **65.7** |
| Neural Cleanse | 94.2 | 94.2 | **98** | **98** | 91.3 | 1 | **94.8** | **94.8** |
| Reconstructive Neuron Pruning | 94.2 | 0 | **97.1** | 0 | 91.3 | 26.3 | **96.5** | **84.9** |
| Feature Shift Tuning | 94.2 | 51.2 | **97.1** | **89** | 91.3 | 93.6 | **97.2** | **98.7** |

According to Table 13, our methods outperform the original attacks when defended by backdoor defense methods in most cases. Defended by Neural Cleanse, the ASR of CTRL drops from 91% to 1%. Optimized by our method, CTRL exhibits 94.8% ASR. What is more, the effectiveness of backdoor defenses primarily hinges on the characteristics of backdoor attacks and backdoor defense methods themselves. For example, SIG fails to penetrate the STRIP (Gao et al. [2019]). In such a case, the attacks optimized by our method also remain futile. Furthermore, our work may benefit Backdoor Defense by considering the distinct importance of samples.

## I    Stealthiness of our components on multiple attacks

In this section, high-quality images of the same category in ImageNet are used to facilitate the comparison between the visibility of various methods.

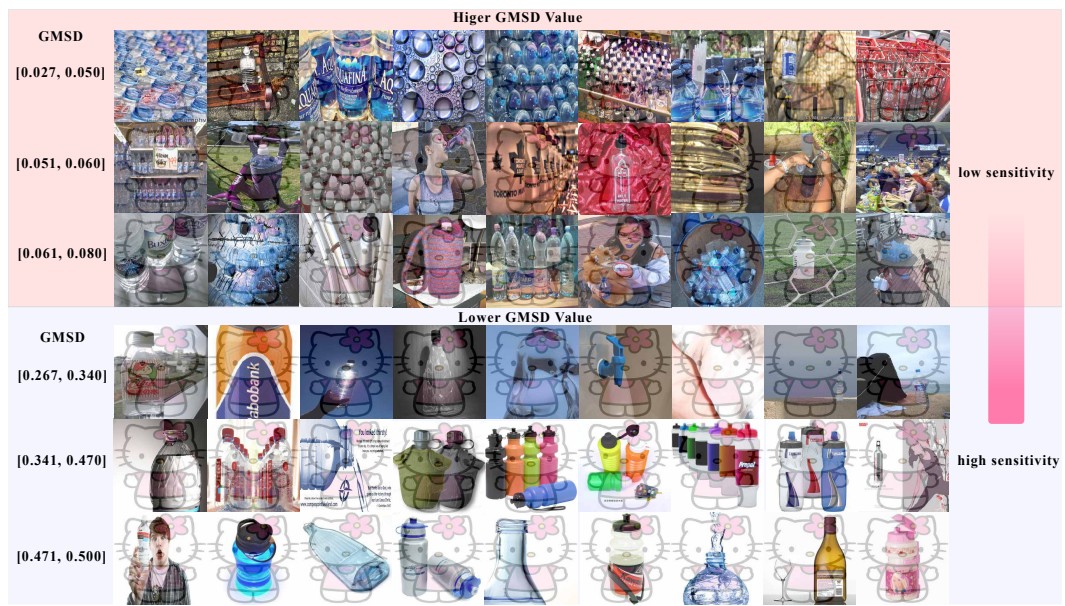

Figure 15: Images poisoned by Blended attacks with different GMSD values.

**The effect of GMSD values in stealthiness enhancement of Blended attacks** As depicted in Figure 15, poisoned images with lower GMSD values exhibit superiority in the stealthiness of triggers for blended attacks. Component B with GMSD tends to find samples with complex backgrounds where the visual sensitivity to Blended triggers will significantly weaken (GMSD $\in [0.027, 0.080]$).In contrast, the Hello Kitty triggers are easy to find when poisoned in images with a simple background (specifically, an all-white patch) where GMSD $\in [0.471, 0.500]$. Therefore, **Component B with GMSD can significantly enhance the stealthiness of Blended attacks.**

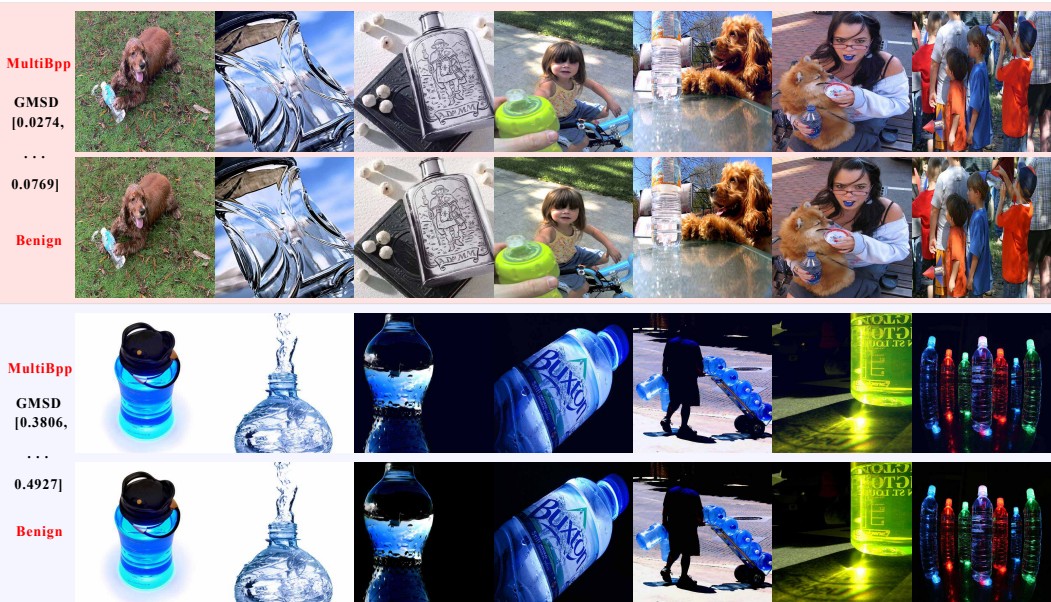

Figure 16: Images poisoned by MultiBpp-B attacks with different GMSD values.

**The effect of GMSD values in stealthiness enhancement of MultiBpp attacks** As depicted in Figure 16, MultiBpp attacks exhibit satisfactory performance in Stealthiness even in the images with the lowest GMSD. Component B with GMSD tends to find samples with complex colors where the visual sensitivity to MultiBpp triggers will significantly weaken (GMSD $\in [0.0274, 0.0769]$). In contrast, single-color dominated images where GMSD $\in [0.3806, 0.4927]$ are selected by Component B to serve as suboptimal samples. The results of lower GMSD suggest that single-channel color variations may amplify susceptibility to MultiBpp attacks under extreme conditions. Specifically,

attenuation of intensity in the green channel (a region of heightened visual sensitivity in the human visual system) and elevation in the blue channel (a region of reduced sensitivity in the human visual system) both result in lower GMSD values. Therefore, single-color dominated samples will not be selected for poisoning. In summary, **Component B with GMSD can significantly enhance the stealthiness of MultiBpp attacks and benefit the performance of Component C**.

## J   Applicability on recent PCBAs and Deployment Cost

We provide a guide to integrate all components with recent PCBAs like Narcissus and Combat. Component A can be simply applied by modifying the poisoning indices. Stronger trigger highlights the Forgetting Events. Triggers with a larger poisoning scope and a larger number of categories in the dataset highlight category diversity. Component B selects samples by comparing the similarity before and after data poisoning, which does not require additional processing. Recent PCBAs typically ensure stealthiness by setting a limit on pixel perturbation thresholds. Component C suffices to apply RGB differentiation processing to these thresholds (e.g., 2:1:3) when training the generator.

What is more, components are intended to be flexibly applied according to the characteristics of the trigger and task requirements. For invisible attacks such as Narcissus, applying components A&C is enough. Thus, we achieve a new SOTA performance in Backdoor Attack based on the SOTA attack (Narcissus). Narcissus achieved a 99% ASR by poisoning 25 images. The ASR of Narcissus drops to 46.11% when we reduce the poisoning rate to 0.00004 (just 2 images). Our methods enhance the ASR from 46.11% to 96.12% with res-log.

Our deployment cost is low. The cost of Component A is the same as the SOTA methods (Forgetting Events), and no training is introduced in Components B or C. Component B requires only a single traversal through the target class (1/10 in CIFAR-10 and 1/100 in CIFAR-100) by maintaining a set with minimal metrics. Component C only requires modification of the poisoning intensity without additional overhead.

