# OpenReview forum: "A Set of Generalized Components to Achieve Effective Poison-only Clean-label Backdoor Attacks with Collaborative Sample Selection and Triggers"
_NeurIPS.cc/2025/Conference — NeurIPS 2025 poster_

### Official Review · Reviewer_CDc3 · 2025-06-25

**Clarity:** 3
**Significance:** 2
**Originality:** 2
**Rating:** 4
**Confidence:** 4

**Summary:**

This paper introduces a set of generalized components aimed at enhancing the effectiveness of Poison-only Clean-label Backdoor Attacks (PCBAs). The core idea revolves around exploring the bidirectional collaborative relationships between sample selection and trigger design. By proposing Component A, Component B, and Component C, the paper seeks to significantly boost both Attack Success Rate (ASR) and stealthiness while maintaining generalization. The work conducts extensive experiments to analyze the impact of these proposed components on various types of backdoor attacks.

**Questions:**

In some tables, the 'no.' column appears to serve only as an experiment identifier. Is there any other practical purpose for this column? If it is merely for numbering, I suggest removing it to improve the tables' conciseness and readability.

**Ethical Concerns:**

["NO or VERY MINOR ethics concerns only"]

**Final Justification:**

I think the overall amount of experiments and the novelty itself are sufficient. I also hope that the defense during the rebuttal discussion can be included as a way to fully examine the attack.

**Limitations:**

Yes.

**Paper Formatting Concerns:**

No.

**Quality:**

3

**Strengths And Weaknesses:**

### Strengths:
- Exploration of ASR Enhancement Mechanisms: Building upon existing research, the paper innovatively explores the collaborative relationship between sample selection (especially by introducing category diversity) and trigger design (e.g., distinct-color poisoning) to improve the ASR of clean-label backdoor attacks. This exploration into the deeper influencing factors of attack mechanisms provides new insights for the field of backdoor attacks.
- Comprehensive Experimental Analysis: The paper conducts a large number of experiments, analyzing how the proposed components (Component A, B, C) impact the ASR and stealthiness of various attack types. This includes analyses of the effects of different negative functions and trigger scales, demonstrating the thoroughness of the research.
### Weaknesses:
- Lack of Backdoor Defense Comparison Experiments: The paper primarily focuses on enhancing backdoor attack effectiveness but completely omits comparative experiments against existing backdoor defense methods. Given that backdoor attacks and defenses are an evolving adversarial process, and numerous mature defense strategies already exist to effectively mitigate backdoor impacts, not considering the attack's performance under practical defense environments constitutes a significant flaw in this paper. Assessing whether these enhanced attacks remain effective in real-world scenarios with active defenses is crucial for evaluating their threat level. To address this, the authors should incorporate experiments comparing their proposed attack against a range of established backdoor defense mechanisms. This should include, but not be limited to, backdoor detection methods such as SCAn [1], and backdoor mitigation/purification methods like RNP [2] and FST [3].
- Inadequate Evaluation of Stealthiness: The paper's evaluation of stealthiness primarily relies on visual demonstrations through images combined with metrics like GMSD. However, the metric and visual presentations alone are insufficient to demonstrate a substantial increase in stealthiness conclusively.

[1] Demon in the variant: Statistical analysis of DNNs for robust backdoor contamination detection

[2] Reconstructive neuron pruning for backdoor defense

[3] Towards stable backdoor purification through feature shift tuning

---

> ### Author Rebuttal · Authors · 2025-07-30
>
> We sincerely appreciate your valuable suggestions and will reply to all concerns in order of importance.
>
> We complete the experiments about **Backdoor Defense** and apply the components to **recent PCBAs**, achieving a **new SOTA performance** in Backdoor Attacks.  The results sufficiently demonstrate the generalization ability and superiority of our method. Related codes and analysis about the new results during the rebuttals will be updated in the paper.
>
> >By poisoning merely **2 images** (poison rate = 2 / 50000 = 0.00004), the optimized Narcissus with res-log as component A achieves **96.12% ASR** and **95.10% BA** in CIFAR-10 with 0 as the target-label.  To the best of our knowledge, **no other attack has comparable performance in ASR with pr = 0.00004 at the clean-label setting.**
>
> # A: Lack of experiments about Backdoor Defense
>
> {Weakness 1}
>
> **{abl: Anti-backdoor learning,ac: activation clustering,fp: Fine-pruning, i-bau: Implicit Hypergradient, nc: Neural Cleanse, np: Reconstructive Neuron Pruning}**
>
> >Given the defense method X, bASR and X-ASR denote the pre-defense and post-defense ASR.
>
> **Badnets**
>
> |method|bASR|abl-ASR|ac-ASR|fp-ASR|i_bau-ASR|nc-ASR|rnp-ASR|
> |:-|:-|:-|:-|:-|:-|:-|:-|
> |random|18.8|0|18|14.4|8.0|18.8|10.5|
> |forget|52.9|8.4|36.3|31|17.3|52.9|8.8|
> |+A|56.2|**14**|47.7|36.5|27.9|56.2|36.3|
> |+B|37.9|3.5|25.6|20.2|**32.5**|37.9|34.9|
> |+C|53.7|5.5|28.4|28.9|7.4|1.0|24.1|
> |+A&C|**87.5**|1.6|**68**|**51.2**|14.8|**81.2**|**47.6**|
> |+B&C|54|0.2|51.8|25.9|5.9|54|0|
>
> **Blend**
>
> |method|bASR|abl-ASR|ac-ASR|fp-ASR|i_bau-ASR|nc-ASR|rnp-ASR|
> |:-|:-|:-|:-|:-|:-|:-|:-|
> |random|57.6|**15.1**|52.4|39.8|28.3|57.6|27.1|
> |forget|76.1|9.2|73.2|63.5|7|76.1|0|
> |+A|77.9|3.9|76.9|64.3|20.0|77.9|36.1|
> |+B|71.7|3.4|67.7|60.1|14.3|71.7|19.7|
> |+C|74.8|6.1|62.9|74.5|48.2|74.8|0|
> |+A&C|**97.1**|1.8|**93.9**|**98.5**|46.1|**96**|**97.3**|
> |+B&C|91|8.9|85.5|92.8|22|91|71.7|
>
> **Other attacks**.
>
> |Attack|Defense|pr|bASR(random)|ASR(random)|bASR(forget)|ASR(forget)|bASR(ours)|ASR(ours)|
> |:-|:-|:-|:-|:-|:-|:-|:-|:-|
> |ctrl|abl|0.03|91.3|66.5|95.7|57.4|96.5|**85.8**|
> |ctrl|ac|0.03|91.3|84.2|95.7|**93.6**|96.5|91|
> |ctrl|fp|0.03|91.3|94.9|95.7|97.4|97.2|**99.2**|
> |ctrl|i-bau|0.03|91.3|39|95.7|**98**|96.5|65.7|
> |ctrl|nc|0.03|91.3|1|95.7|**95.7**|94.8|94.8|
> |ctrl|rnp|0.03|91.3|26.3|95.7|42.2|96.5|**84.9**|
> |sig|ac|0.03|94.2|93.5|96.5|96.4|97.2|**97.5**|
> |sig|fp|0.03|94.2|61.6|96.5|75.8|98|**88.4**|
> |sig|i-bau|0.03|94.2|8.9|96.5|17.7|97.2|**42.4**|
> |sig|nc|0.03|94.2|94.2|96.5|96.5|98|**98**|
> >The design of A&B depends on the specific requirements for concealment and ASR. The following results adequately address the concerns as the expected results of A&B are between \[B, A\].
> >Optimal improvement in ASR cannot be achieved when consideration is given to stealthiness. Therefore, components A&C exhibit the best ASR improvement.
>
> ## A.1 Our methods **outperform** the original attacks when defended by backdoor defense methods.
> >Defended by nc, the ASR of CTRL drops from 91% to 1%. Optimized by our method, nc exhibits 94.8% ASR.
>
> ## A.2 The effectiveness of backdoor defenses primarily hinges on **the characteristics of backdoor attacks themselves**.
> >sig fails to penetrate the STRIP. In such a case, the attacks optimized by our method also remain futile.
>
> ## A.3 Our work may **benefit** Backdoor Defense by considering the distinct importance of samples in backdoor defense.
>
> # B: Stealthiness Evaluation
>
> {Weakness2}
>
> ## B.1 **Visual presentation of poisoned images** selected by Component B with different MSD and GMSD values is provided in **Appendix I**.
>
> ## B.2 Your concerns are entirely justified, but this stems from **inherent flaws in the evaluation metric design** rather than flaws of Component B.
> >Component B is designed for machine-based applications. For any machine-quantifiable evaluation metric provided, it can rapidly identify images that achieve optimal performance under that metric. Given a fixed trigger, component B selects images with optimal stealthiness rather than making a visible trigger totally invisible.
>
> ## B.3 Component B is the pioneering work to enhance stealthiness via sample selection, which is **universally applicable** to both machine and human evaluations.
>
> ## B.4 **Simplicity is not equal to the lack of novelty.** The **difference** between the design of plug-and-play components and other works may potentially lead to misinterpretations where our advantage is misconstrued as a limitation.
> >Achieving simplicity while maintaining efficiency is a core design principle for plug-and-play components for strong applicability and low computational overhead.
>
> >Unlike other works simply being demanded to train a powerful model to use, we need to design components that can easily be applied to most attacks at the code level.
>
> ## B.5 We select the advanced metric to ensure the evaluation of similarity.
> >GMSD demonstrates higher sensitivity to structural distortions (such as blurring and compression artifacts) than SSIM. Additionally, SSIM suffers from instability in white noise assessment and exhibits deviations from subjective scores under specific distortion conditions. In contrast, GMSD shows stronger consistency with human subjective evaluations.
>
> # C: Advice for the paper
> Due to space constraints, no. are employed here to facilitate textual analysis in the paper. In the formal version, where space limitations are alleviated, the presentation format will be refined accordingly.
>
> # D: Our work is really **valuable**
> ##  We achieve a **new SOTA performance** in Backdoor Attack by merely using component A based on the SOTA attack (Narcissus).
> >By poisoning merely **2 images** (poison rate = 2 / 50000 = 0.00004), Narcissus with res-log achieves **96.12% ASR** and **95.10% BA** in CIFAR-10 with 0 as the target-label.
>
> >In other settings, we follow the default settings in the official code repository. ASRs and BAs are calculated by **the mean of ASR** between epochs {180-200}.
>
> |Methods|ASR|BA|
> |:-|:-|:-|
> |Narcissus|46.11|94.79|
> |ours|**96.12**|**95.10**|
> >In the offical paper, Narcissus achieved a 99% ASR by poisoning **25 images**.  The ASR of Narcissus **drops to 46.11%** when we reduce the poisoning rate to 0.00004 (just 2 images). Our method enhance the ASR **from 46.11% to 96.12%** with res-log as component A.
>
> We hope our response can help resolve your confusion, and we will continue to optimize the parts that may cause confusion.

---

> > ### Comment · Reviewer_CDc3 · 2025-08-04
> > **Response**
> >
> > I think these points are sufficient, so I'm willing to improve my score. However, I still have a few questions. First, you didn't implement all of the backdoor defense methods I recommended. As far as I know, there should be existing code libraries for these methods, and I'd like to know why. Second, I strongly agree with what you said about the role of component B. My original intention was to use common SSIM or PSNR metrics to illustrate your stealth improvements. Or am I misunderstanding that GMSD itself is a way to measure stealth?

---

> ### Author Response · Authors · 2025-08-04
> **Response to new questions**
>
> Thank you so much for the reply. We are delighted to optimize our paper based on the sustaining advice.
> # A: Answer to Question 1
>
> ## A1 Our experiments are based on **BackdoorBench**.
> > - After receiving the reviews, we recognized the reviewers' concern on the effect of **Sample Selection** in **Backdoor Defense**. We adopted the **BackdoorBench**, a comprehensive benchmark of backdoor learning, to provide sufficient experiments upon **mainstream backdoor attack and defense methods** within just **a few days**. Among the defense methods you suggested, we only found the support for **RNP** in **Supported defenses** of **BackdoorBench**.
>
> ## A2 **Instead of** proposing new attack methods to penetrate all defenses, we propose generalized components to improve both **ASR and stealthiness** of attacks.
> >- Therefore, we focus on addressing the concerns about whether our methods would **negatively impact** the existing attack capabilities to penetrate defenses at most cases. **It is unnecessary and impractical to exhaustively enumerate all defense methods.**
> >- The enhanced penetration effectiveness of our methods upon backdoor defense is an additional priority thanks to the valuable suggestions from reviewers.
>
> ## A3 Our experiments are **sufficient** to explore the effect of our components on Backdoor Defense.
>
> >- During the **Rebuttal**, we apply our methods to different attacks and defense methods in different settings (**4** attack methods, **6** defense methods, **6+1** sample selection methods (**random, loss, grad, forget, component A, component B**) + **Component C**, **: 4 X 6 X 7 = 168 settings**). As for Badnets and Blend, we need to explore the effect of components (different combinations of components ) in Defense by designing more settings.
>
> >-  Experiments to explore the effect of our components upon defense are **not** tightly related to specific defense methods. The experiments we have conducted cover a sufficient number of backdoor defense methods with **comparable defensive capabilities**. All related codes and logs during the training will be released.
>
> ## A4 However, to better address the reviewers' concerns, we are cooking the experiments in the left limited time to provide more perfect experiment evaluation.
>
> >- We **provide** the effect of our methods on the proposed defense methods (**Scan and FST**).
>
> **FST**
> |Attack|Defense|pr|bASR(random)|ASR(random)|bASR(forget)|ASR(forget)|bASR(ours)|ASR(ours)|
> |:-|:-|:-|:-|:-|:-|:-|:-|:-|
> |ctrl|fst|0.03|91.3 | 93.6 | 95.7 |88.7  | 97.2 |**98.7** |
> |sig |fst|0.03|94.2 |51.2  |96.5 |86.9  | 97.1 | **89**|
> |badnets|fst|0.03|18.8|22|52.9|54.7|87.5|**86.4**|
> |blend|fst|0.03|57.6|42.8|76.1|64.1|97.1|**90.9**|
>
> **SCAn**
> |Attack|method|pr|bASR(random)|TPR(random)|bASR(forget)|TPR(forget)|bASR(ours)|TPR(ours)|
> |:-|:-|:-|:-|:-|:-|:-|:-|:-|
> |ctrl|SCAn|0.03 |91.3  |94.5  |95.7  |97.7  | 94.8 | 96.4 |
> |sig|SCAn|0.03 | 94.2 |98  |96.5  | 98 | 97.1 | 97.3 |
> |badnets|SCAn|0.03 | 18.8 |81.5|52.9  |89.2  | 54.3 | 87.4 |
> |blend|SCAn|0.03 | 57.6 | 91.3 | 76.1 | 96 | 74.9 |  95.1|
>
> >- According to the results, we can conclude that our methods enhance or keep the performance of attacks in defense methods. The results will be jointly reported with other experiment results in the future paper version.
>
> # B: Answer to Question 2
>
> Yes, **GMSD** is utilized to measure the similarity between images.
> >- During the preliminary exploration phase of our work, we **compared images selected based on different metrics**.
>
> >- The images chosen based on **SSIM** do not meet our expectations, as there is a **discrepancy** between the metric results and subjective judgments.
>
> >- In contrast, the images selected based on **GMSD** exhibit significantly better subjective perceptions of stealthiness compared to those chosen using SSIM. Therefore, we ultimately select **GMSD** as a **representative metric**.

---

> ### Author Response · Authors · 2025-08-05
> **Experiments on proposed defense methods have been updated**
>
> Hello,
>
>   The experiments on requested dense methods (**Scan and FST**) have been **updated** in the **Response to new questions**.
>
>   We hope our response can help resolve your confusion, and we will continue to optimize the parts that may cause confusion.

---

> ### Comment · Reviewer_5WGK · 2025-08-05
>
> I want to thank the authors for their response. The authors addressed my concerns. Also, I read the comments and follow-up questions of other reviewers, and I believe the authors also answered these questions sufficiently and promptly. Therefore, I will improve my rating.

---

> > ### Author Response · Authors · 2025-08-05
> > **Thanks for the advice**
> >
> > We are deeply grateful for your assistance and recognition of our work. It has been a pleasure to address your concerns.

---

> ### Author Response · Authors · 2025-08-05
> **Thanks for the advice**
>
> We are deeply grateful for your assistance and recognition of our work. It has been a pleasure to address your concerns.

---

### Official Review · Reviewer_5WGK · 2025-07-01

**Clarity:** 2
**Significance:** 2
**Originality:** 2
**Rating:** 4
**Confidence:** 4

**Summary:**

This paper proposes a novel method, using trigger scales and features to select better ones, and leverages RGB channels to enhance the trigger’s backdoor attack performance.

**Questions:**

- Could you please explain why the attack performance degrades when Component B is added (e.g., as shown in section 3.1)?

**Ethical Concerns:**

["NO or VERY MINOR ethics concerns only"]

**Final Justification:**

The authors addressed my concerns and other reviewers' questions.

**Limitations:**

Yes.

**Quality:**

3

**Strengths And Weaknesses:**

Strengths:
- This paper provides a brand new perspective to enhance image trigger using RGB channels, which can also decrease the interference with the original trigger.
- The authors conducted multiple experiments to show the performance of their methods, they also carried out several ablation studies.

Weaknesses:
- Component A seems to be overweighted, which doesn’t show much technical contribution and innovation, as it is simply based on the selection of trigger scales.
- Some claims would be more convincing with proper evaluations and validations. For examples, the authors claim that the proposed method can bypass human inspection. It would be worthy providing quantitively comparison between original images and perturbation samples for Component B. MSE or SSIM would be good metrics for that.
- The correlation between design choice and the attack performance remains unclear. For example, the combination of Component A and C yields the best performance. Is it an empirical observation or what is the rationale behind this observation? As the main contribution of the paper is the design choices, it would be ideal to elaborate the concreted impact of each component on the attack performance.

---

> ### Author Rebuttal · Authors · 2025-07-30
>
> We sincerely appreciate your valuable suggestions and will reply to all concerns in order of importance.
>
> We complete the experiments about **Backdoor Defense** and apply the components to **recent PCBAs**, achieving a **new SOTA performance** in Backdoor Attacks. Related codes and analysis about the new results during the rebuttals will be updated in the paper.
>
> # A Concerns about Simplicity and Novelty
>
> {Weakness1}
>
> ## A.1 We achieve a **new SOTA performance** in Backdoor Attack by merely using component A based on the SOTA attack (Narcissus).
> >By poisoning merely **2 images** (poison rate = 2 / 50000 = 0.00004), Narcissus with res-log achieves **96.12% ASR** and **95.10% BA** in CIFAR-10 with 0 as the target-label.
>
> >In other settings, we follow the default settings in the official code repository. ASRs and BAs are calculated by **the mean of ASR** between epochs {180-200}.
>
> |Methods|ASR|BA|
> |:-|:-|:-|
> |Narcissus|46.11|94.79|
> |ours|**96.12**|**95.10**|
> >In the offical paper, Narcissus achieved a 99% ASR by poisoning **25 images**.  The ASR of Narcissus **drops to 46.11%** when we reduce the poisoning rate to 0.00004 (just 2 images). Our method enhance the ASR **from 46.11% to 96.12%** with res-log as component A. To the best of our knowledge, **no other attack has comparable performance in ASR with pr = 0.00004 at the clean-label setting.**
>
> ## A.2 Simplicity is not equal to the lack of novelty. The **difference between the design of plug-and-play components and other works** may potentially lead to misinterpretations where our advantage is misconstrued as a limitation.
> >Achieving simplicity while maintaining efficiency is a core design principle for plug-and-play components for strong applicability and low computational overhead.
>
> >Unlike other works simply being demanded to train a powerful model to use, we need to design components that can easily be applied to most attacks at the code level.
>
> ## A.3 We relocated a **significant portion** of the theoretical work and ablation study of Component B&C in the **Appendix**. We will adjust our paper for better readability.
> >In total, we dedicate **Our methods** and **Appendix {B, C, D}** to establish the theoretical foundation of the components. Additionally, background knowledge of the human visual system related to Component C is provided in **Appendix E**.
>
> >Furthermore, detailed theoretical explanations and thorough ablation experiments are elaborated in **Appendices {D, E}** and **Appendix {I, F}**.
>
> ## A.4 **The innovations are outlined as follows:**
> >**We are the first to investigate the interplay between Trigger Design and Sample Selection in backdoor attacks.**
>
> >**We are the first to focus on universal component design to strengthen generalization across all backdoor attacks.**
>
> >**We are the first to introduce biomedical foundations to optimize trigger design in this domain.**
>
> ## A.5 Model interpretability itself has long been a significant challenge in the field of AI.
> >The mathematical interpretation of Component A needs to encompass the mathematical relationship between changes in inputs (e.g., triggers and trigger variations) and corresponding changes in outputs of DNNs. **A precise mathematical underpinning of the irrelation between inputs and outputs of DNNs is currently impractical.**
>
> >Therefore, the mathematical sections in most papers of Backdoor Attack focus on algorithm elaboration for trigger generation and modeling of Backdoor Attack objectives.
>
> ## A.6 Macroscopic analysis of the effect of the poisoning area is **provided** in the paper.
> >Models with triggers covering larger poisoning regions require analysis of more pixels in the corresponding areas for classification, and thus are more susceptible to being interfered with by features of other classes. Samples with larger category diversity are inherently "cross-class," which mitigates the interference.
>
> ## A.7 As for the issue of **proportions**, components B&C represent the pioneering works with **no methods for comparison**.
>
> # B Stealthiness Evaluation
>
> {Weakness2}
>
> ## B.1 **Visual presentation of poisoned images** selected by Component B with different MSD and GMSD values is provided in **Appendix I**.
>
> ## B.2 Your concerns are entirely justified, but this stems from inherent flaws in the evaluation metric design rather than flaws of Component B.
> >Component B is designed for machine-based applications. For any machine-quantifiable evaluation metric provided, it can rapidly identify images that achieve optimal performance under that metric. **Given a fixed trigger, component B selects images with optimal stealthiness rather than making a visible trigger totally invisible.**
>
> ## B.3 Component B is the pioneering work to enhance stealthiness via sample selection, which is **universally applicable** to both machine and human evaluations.
>
> ## B.4 Background knowledge on the human visual system related to Component C is **provided** in **Appendix E**.
>
> ## B.5 We select the advanced metric to ensure the evaluation of similarity.
> >GMSD demonstrates higher sensitivity to structural distortions (such as blurring and compression artifacts) than SSIM. Additionally, SSIM suffers from instability in white noise assessment and exhibits deviations from subjective scores under specific distortion conditions. In contrast, GMSD shows stronger consistency with human subjective evaluations.
>
> # C Questions about Results
>
> {Weakness3,Question1}
>
> ## C.1 Each component is designed with **distinct objectives**.
> >Component B serves as a sample selection method aimed at enhancing stealthiness, whereas Component A is designed to improve the asr.
> >Component C is a reasonable adjustment for better ASR and stealthiness.
>
> ## C.2 **Optimal improvement in ASR cannot be achieved when consideration is given to stealthiness.** Therefore, components A&C without B exhibit the best ASR improvement.
>
> ## C.3 Our components are not intended to be used in a bound manner. Adopting a unified objective function and framework for handling is **inflexible**.
> >Given the drawbacks of different scenarios and triggers, it is necessary to allow attackers to freely control the proportion of stealthiness and ASR by utilizing the components either individually or in combination.
>
> >For an invisible Narcrssus attack where the trigger design inherently considers perturbation intensity, Component B is unnecessary, and directly employing Components A constitutes the new SOTA performance.
>
> ## D Backdoor Defense
>
> **{abl: Anti-backdoor learning,ac: activation clustering,fp: Fine-pruning, i-bau: Implicit Hypergradient, nc: Neural Cleanse, np: Reconstructive Neuron Pruning}**
> >Given the defense method X, bASR and X-ASR denote the pre-defense and post-defense ASR.
>
> **Badnets**
>
> |method|bASR|abl-ASR|ac-ASR|fp-ASR|i_bau-ASR|nc-ASR|rnp-ASR|
> |:-|:-|:-|:-|:-|:-|:-|:-|
> |random|18.8|0|18|14.4|8.0|18.8|10.5|
> |forget|52.9|8.4|36.3|31|17.3|52.9|8.8|
> |+A|56.2|**14**|47.7|36.5|27.9|56.2|36.3|
> |+B|37.9|3.5|25.6|20.2|**32.5**|37.9|34.9|
> |+C|53.7|5.5|28.4|28.9|7.4|1.0|24.1|
> |+A&C|**87.5**|1.6|**68**|**51.2**|14.8|**81.2**|**47.6**|
> |+B&C|54|0.2|51.8|25.9|5.9|54|0|
>
> **Blend**
>
> |method|bASR|abl-ASR|ac-ASR|fp-ASR|i_bau-ASR|nc-ASR|rnp-ASR|
> |:-|:-|:-|:-|:-|:-|:-|:-|
> |random|57.6|**15.1**|52.4|39.8|28.3|57.6|27.1|
> |forget|76.1|9.2|73.2|63.5|7|76.1|0|
> |+A|77.9|3.9|76.9|64.3|20.0|77.9|36.1|
> |+B|71.7|3.4|67.7|60.1|14.3|71.7|19.7|
> |+C|74.8|6.1|62.9|74.5|48.2|74.8|0|
> |+A&C|**97.1**|1.8|**93.9**|**98.5**|46.1|**96**|**97.3**|
> |+B&C|91|8.9|85.5|92.8|22|91|71.7|
> >The design of A&B depends on the specific requirements for concealment and ASR. The following results adequately address the concerns as the expected results of A&B are between \[B, A\].
> >Optimal improvement in ASR cannot be achieved when consideration is given to stealthiness. Therefore, components A&C exhibit the best ASR improvement.
>
> **Other attacks**.
>
> |Attack|Defense|pr|bASR(random)|ASR(random)|bASR(forget)|ASR(forget)|bASR(ours)|ASR(ours)|
> |:-|:-|:-|:-|:-|:-|:-|:-|:-|
> |ctrl|abl|0.03|91.3|66.5|95.7|57.4|96.5|**85.8**|
> |ctrl|ac|0.03|91.3|84.2|95.7|**93.6**|96.5|91|
> |ctrl|fp|0.03|91.3|94.9|95.7|97.4|97.2|**99.2**|
> |ctrl|i-bau|0.03|91.3|39|95.7|**98**|96.5|65.7|
> |ctrl|nc|0.03|91.3|1|95.7|**95.7**|94.8|94.8|
> |ctrl|rnp|0.03|91.3|26.3|95.7|42.2|96.5|**84.9**|
> |sig|ac|0.03|94.2|93.5|96.5|96.4|97.2|**97.5**|
> |sig|fp|0.03|94.2|61.6|96.5|75.8|98|**88.4**|
> |sig|i-bau|0.03|94.2|8.9|96.5|17.7|97.2|**42.4**|
> |sig|nc|0.03|94.2|94.2|96.5|96.5|98|**98**|
>
> ## D.1 Our methods **outperform** the original attacks when defended by backdoor defense methods.
> >Defended by nc, the ASR of CTRL drops from 91% to 1%. Optimized by our method, nc exhibits 94.8% ASR.
>
> ## D.2 The effectiveness of backdoor defenses primarily hinges on the characteristics of backdoor attacks themselves.
> >sig fails to penetrate the STRIP. In such a case, the attacks optimized by our method also remain futile.
>
> ## D.3 Our work may **benefit** Backdoor Defense by considering the distinct importance of samples into consideration.
>
> We hope our response can help resolve your confusion, and we will continue to optimize the parts that may cause confusion.

---

> > ### Comment · Reviewer_5WGK · 2025-08-07
> >
> > Thank you for the detailed response. All my major concerns were addressed. I also read the interactive communication between the authors and other reviewers, and I saw that many other issues and questions were resolved. I believe this paper brings some insights and new knowledge to our community.

---

### Official Review · Reviewer_AdY7 · 2025-07-02

**Clarity:** 2
**Significance:** 2
**Originality:** 2
**Rating:** 4
**Confidence:** 3

**Summary:**

This paper proposed a framework for improving poison-only clean-label backdoor attacks (PCBAs) by introducing a set of generalized components that jointly optimize sample selection and trigger design. Specifically, the paper introduces three core components: A, B and C. The paper demonstrate that these components can be flexibly combined to improve a wide range of existing PCBA methods (e.g., Badnets, Blended, BppAttack) across CIFAR-10 and CIFAR-100 datasets.

**Questions:**

1) The current component integration strategy relies on heuristic rules and thresholds. Is there a way to formalize the combination of Components A and B under a joint optimization objective?
2) How does your framework perform under standard backdoor defense methods, such as STRIP, Neural Cleanse, or Fine-Pruning? Since your method emphasizes stealthiness, it would be especially valuable to show how the selected samples and RGB-optimized triggers evade detection by these defenses.
3) While the paper explores pairwise combinations, a more in-depth analysis of why certain combinations (e.g., A+B vs. A+C) lead to better or worse ASR would be useful.  What are the interactions (e.g., synergistic or conflicting effects) between the three components, and how do they influence each other’s effectiveness?
4) Since Component C exploits perceptual RGB sensitivity, it would be valuable to include a perceptual user study or visibility score (e.g., via SSIM or psychometric testing) to support the stealthiness claim from a human inspection perspective.

**Ethical Concerns:**

["NO or VERY MINOR ethics concerns only"]

**Final Justification:**

The rebuttal address most of my concerns. I will raise my score.

**Limitations:**

See Questions and Weaknesses.

**Quality:**

2

**Strengths And Weaknesses:**

Strengths:
1) The paper introduced a two-way collaborative mechanism of sample selection and trigger design, which is different from the previous studies that treated the two in isolation. In the optimization of trigger concealment, the mechanism of the difference in the sensitivity of the human visual system to RGB was introduced for the channel reallocation design (Component C) for the first time, which is novel and practical.
2) It has strong adaptability to scenarios with low poisoning rates and practical value in actual attack applications.
3) The method can be used in various model structures (such as ResNet-18, VGG16, DenseNet121), it has strong generalization.

Weaknesses:
1) The application scope is limited to the clean label scenario.
2) There is a lack of a unified theoretical framework or optimization objective function support among the three components. The combination of Component A and B is based on simple sorting and threshold sampling, lacking a more systematic optimization strategy.
3) Resistance to existing defense methods such as STRIP, Fine-Pruning, Neural Cleanse was not analyzed.
4) The experiments present conflicting results. Table 1 shows that adding Component B can lead to a decrease in ASR (e.g., for Badnets-C at 1% poisoning rate, ASR drops from 86.15% with Component A&C to 77.67% with Components A&B&C). In addition to that, in some cases, the BA also declines with Component B's inclusion.

---

> ### Author Rebuttal · Authors · 2025-07-30
>
> We sincerely appreciate your valuable suggestions and will reply to all concerns in order of importance.
>
> We complete the experiments about **Backdoor Defense** and apply the components to **recent PCBAs**, achieving a **new SOTA performance** in Backdoor Attacks.  Related codes and analysis about the new results during the rebuttals will be updated in the paper.
>
> # A Lack of Defense Experiments
>
> {Weakness3, Question2}
>
> **{abl: Anti-backdoor learning,ac: activation clustering,fp: Fine-pruning, i-bau: Implicit Hypergradient, nc: Neural Cleanse, np: Reconstructive Neuron Pruning, STRIP}**
>
> >Given the defense method X, bASR and X-ASR denote the pre-defense and post-defense ASR.
>
> **Badnets**
>
> |method|bASR|abl-ASR|ac-ASR|fp-ASR|i_bau-ASR|nc-ASR|rnp-ASR|
> |:-|:-|:-|:-|:-|:-|:-|:-|
> |random|18.8|0|18|14.4|8.0|18.8|10.5|
> |forget|52.9|8.4|36.3|31|17.3|52.9|8.8|
> |+A|56.2|**14**|47.7|36.5|27.9|56.2|36.3|
> |+B|37.9|3.5|25.6|20.2|**32.5**|37.9|34.9|
> |+C|53.7|5.5|28.4|28.9|7.4|1.0|24.1|
> |+A&C|**87.5**|1.6|**68**|**51.2**|14.8|**81.2**|**47.6**|
> |+B&C|54|0.2|51.8|25.9|5.9|54|0|
>
> **Blend**
>
> |method|bASR|abl-ASR|ac-ASR|fp-ASR|i_bau-ASR|nc-ASR|rnp-ASR|
> |:-|:-|:-|:-|:-|:-|:-|:-|
> |random|57.6|**15.1**|52.4|39.8|28.3|57.6|27.1|
> |forget|76.1|9.2|73.2|63.5|7|76.1|0|
> |+A|77.9|3.9|76.9|64.3|20.0|77.9|36.1|
> |+B|71.7|3.4|67.7|60.1|14.3|71.7|19.7|
> |+C|74.8|6.1|62.9|74.5|48.2|74.8|0|
> |+A&C|**97.1**|1.8|**93.9**|**98.5**|46.1|**96**|**97.3**|
> |+B&C|91|8.9|85.5|92.8|22|91|71.7|
>
> >Optimal improvement in ASR cannot be achieved when consideration is given to stealthiness. Therefore, components A&C exhibit the best ASR improvement.
>
> >The design of A&B depends on the specific requirements for concealment and ASR. The following results adequately address the concerns as the expected results of A&B are between \[B, A\].
>
> We also provide the results of other attacks with defense methods.
>
> |Attack|Defense|pr|bASR(random)|ASR(random)|bASR(forget)|ASR(forget)|bASR(ours)|ASR(ours)|
> |:-|:-|:-|:-|:-|:-|:-|:-|:-|
> |ctrl|abl|0.03|91.3|66.5|95.7|57.4|96.5|**85.8**|
> |ctrl|ac|0.03|91.3|84.2|95.7|**93.6**|96.5|91|
> |ctrl|fp|0.03|91.3|94.9|95.7|97.4|97.2|**99.2**|
> |ctrl|i-bau|0.03|91.3|39|95.7|**98**|96.5|65.7|
> |ctrl|nc|0.03|91.3|1|95.7|**95.7**|94.8|94.8|
> |ctrl|rnp|0.03|91.3|26.3|95.7|42.2|96.5|**84.9**|
> |sig|ac|0.03|94.2|93.5|96.5|96.4|97.2|**97.5**|
> |sig|fp|0.03|94.2|61.6|96.5|75.8|98|**88.4**|
> |sig|i-bau|0.03|94.2|8.9|96.5|17.7|97.2|**42.4**|
> |sig|nc|0.03|94.2|94.2|96.5|96.5|98|**98**|
>
> |Attack|method|pr|bASR(random)|TPR(random)|bASR(forget)|TPR(forget)|bASR(ours)|TPR(ours)|
> |:-|:-|:-|:-|:-|:-|:-|:-|:-|
> |ctrl|STRIP|0.03|91.3|51.9|95.7|52|96.5|44.2|
> |sig|STRIP|0.03|94.2|82.4|96.5|93.8|98|95.7|
> |badnets|STRIP|0.03|18.8|6.7|52.9|69.8|87.5|14.9|
> |blend|STRIP|0.03|57.6|1.1|76.1|2.4|91|1.9|
>
> ## A.1 Our methods **outperform** the original attacks when defended by backdoor defense methods in most cases.
> >Defended by nc, the ASR of CTRL drops from 91% to 1%. Optimized by our method, nc exhibits 94.8% ASR.
>
> ## A.2 The effectiveness of backdoor defenses primarily hinges on the characteristics of backdoor attacks themselves.
> >sig fails to penetrate the STRIP. In such a case, the attacks optimized by our method also remain futile.
>
> ## A.3 Our work may benefit Backdoor Defense by considering the distinct importance of samples.
>
> # B Questions about ASR
>
> {Weakness4,Question3}
>
> ## B.1 Optimal improvement in ASR cannot be achieved when consideration is given to stealthiness.
> >Component B serves as a sample selection method aimed at **enhancing stealthiness**, whereas Component A is designed to improve the asr. Component C is a reasonable adjustment for better ASR and stealthiness.
>
> >Component B eliminates some images that contribute to improving the ASR  but are detrimental to stealthiness. Therefore, components A&C exhibit the best ASR improvement.
>
> ## B.2 Components are intended to be flexibly adjusted according to the characteristics of the trigger and task requirements.
> >For invisible attacks such as Narcissus, applying components A&C (even only A) is enough. We achieve a **new SOTA performance** based on the SOTA attack (Narcissus).
> >>By poisoning merely **2 images (poison rate = 0.00004)**, Narcissus with res-log achieves 96.12% ASR and 95.10% BA in CIFAR-10 with 0 as the target-label. We follow the default settings in the official code repository. ASRs and BAs are the mean ASR between epochs {180-200}.
>
> |Methods|ASR|BA|
> |:-|:-|:-|
> |Narcissus|46.11|94.79|
> |ours|**96.12**|**95.10**|
>
> ## B.3 Compared to benign BA results (94%), the decrease of BA within 0.5% in backdoor attacks (min: 93.70%, max: 94.22%) is a normal and imperceptible training perturbation of accuracy.
>
> # C Concerns about Dirty-label Poisoning
>
> {Weakness1}
>
> ## C.1 Remediation to the dirty label is **unnecessary** in our paper.
> >We aim to optimize dirty-label attacks to clean-label attacks while preserving high ASR, rendering further optimization in dirty-label scenarios less critical in our paper.
>
> ## C.2 We provide an analysis of the phenomenon.
> >Under clean-label settings (e.g., take airplane as the target label), component A selects images of airplanes that least resemble airplanes based on the forgetting events with its category diversity. According to the phenomenon of models taking shortcuts discussed in another paper, models tend to rely on learning triggers as a shortcut to solve the hard task in the selected images.  Therefore, component A gets higher ASRs because generic airplane features exert minimal interference from backdoor features.
>
> >In contrast, under dirty-label settings, it is more effective to directly use cat images for poisoning instead of selecting airplane images that least resemble airplanes. The model faces the greatest difficulty in classifying cats as airplanes and resorts to backdoor shortcuts, resulting in higher ASRs.
>
> # D Stealthiness Evaluation
>
> {Question4}
>
> ## D.1 **Visual presentation of poisoned images selected by Component B** with different MSD and GMSD values is provided in **Appendix I**.
> >GMSD demonstrates higher sensitivity to structural distortions (such as blurring and compression artifacts) than SSIM. Additionally, SSIM suffers from instability in white noise assessment and exhibits deviations from subjective scores under specific distortion conditions. In contrast, GMSD shows stronger consistency with human subjective evaluations.
>
> ## D.2 Your concerns are entirely justified, but this stems from inherent flaws in the evaluation metric design rather than flaws of Component B.
> >Component B is designed for machine-based applications. For any machine-quantifiable evaluation metric provided, it can rapidly identify images that achieve optimal performance under that metric. Given a fixed trigger, component B selects images with optimal stealthiness rather than making a visible trigger totally invisible.
>
> ## D.3 Component B is the pioneering work to enhance stealthiness via sample selection, which is universally applicable to both machine and human evaluations.
>
> # E Theoretical Concerns
>
> {Weakness2,Question1}
>
> ## E.1 We relocate a significant portion of the theoretical work to the Appendix.
> >In total, we dedicate **Our methods** and **Appendix {B, C, D}** to establish the theoretical foundation of the components. Additionally, background knowledge of the human visual system related to Component C is provided in **Appendix E**.
>
> ## E.2 Unlike other work being demanded to train a powerful model to use, we need to design components that can easily be applied to most attacks at the code level.
> >Achieving simplicity while maintaining efficiency is a core design principle for plug-and-play components for strong applicability and low computational overhead.
>
> >The difference may potentially lead to misinterpretations where our advantage is misconstrued as a limitation.
>
> >The integration among components does not require neural networks, thus eliminating the need for objective function design. Once plug-and-play components are incorporated into neural networks, it involves integration with the target attack, leading to increased conflicts at the code level and hindering rapid application.
>
> ## E.3 Stealthiness and ASR represent independent dimensions, and the relative importance of their proportions cannot be quantitatively compared.
>
> ## E.4 Designing an automated framework is akin to learning rate searching algorithms to find optimal values.
>
> ## E.5 Model interpretability itself has long been a significant challenge in the field of AI. A precise mathematical underpinning of the irrelation between inputs and outputs of DNNs is currently impractical.
> >The mathematical interpretation of Component A needs to encompass the mathematical relationship between changes in inputs (e.g., triggers and trigger variations) and corresponding changes in outputs of DNNs. A precise mathematical underpinning of the irrelation between inputs and outputs of DNNs is currently impractical.Therefore, **the mathematical sections in most papers of Backdoor Attack focus on algorithm elaboration for trigger generation and modeling of Backdoor Attack objectives**.
>
> ## E.6 Macroscopic analysis of the effect of the poisoning area is provided in the paper.
> >Models with triggers covering larger poisoning regions require analysis of more pixels in the corresponding areas for classification, and thus are more susceptible to being interfered with by features of other classes. Samples with larger category diversity are inherently "cross-class," which mitigates the interference.
>
> ## E.7 Our work aims to motivate the community to prioritize and investigate this new direction.
> >As a pioneering approach, establishing a comprehensive theory on the collaboration between sample selection and triggers is a long-term and arduous task.
>
> We hope our response can help resolve your confusion, and we will continue to optimize the parts that may cause confusion.

---

> > ### Comment · Reviewer_AdY7 · 2025-08-07
> >
> > Thank you for the rebuttal with additional experiments. I understand it is challenging to conduct them in such a short time. In general, the rebuttal addressed most of my comments, thus I will elevate my scores accordingly.

---

> > > ### Author Response · Authors · 2025-08-08
> > > **Thanks for the advice**
> > >
> > > We are deeply grateful for your assistance and recognition of our work. It has been a pleasure to address your concerns.

---

> ### Author Response · Authors · 2025-08-05
>
> Hello!
>
> We would like to know if there are some concerns there, and we are really willing to continuously improve our work based on valuable suggestions.

---

> ### Comment · Area_Chair_Td53 · 2025-08-07
> **Please Read Author Response and Share Your Post-Rebuttal Comments**
>
> Dear Reviewer AdY7,
>
> Thank you for your thoughtful and constructive reviews of this submission. Your time and expertise are greatly appreciated.
>
> The author response is now available. Please carefully read the rebuttal and update your reviews with post-rebuttal comments, indicating whether the response adequately addresses your concerns and whether it changes your assessment of the paper.
>
> Please also feel free to engage in discussion with your fellow reviewers, especially in cases where there are divergent opinions, so we can reach a more accurate and well-informed consensus. If you have any points of disagreement or clarification, this is a good time to raise and explore them collaboratively.
>
> Best regards,
>
> AC

---

### Official Review · Reviewer_CWuV · 2025-07-03

**Clarity:** 3
**Significance:** 3
**Originality:** 3
**Rating:** 5
**Confidence:** 4

**Summary:**

This work introduces a modular framework of three “generalized components” that jointly optimize sample selection and trigger design to boost the success rate and stealth of clean-label backdoor attacks. Component A selects hard-to-learn images via a balance of forgetting events and category diversity; Component B uses visual-quality metrics to hide triggers in images; Component C reallocates poisoning strength across RGB channels according to human perceptual sensitivity. Integrated into various backdoor attacks, these components yield 20–60 pp ASR gains on CIFAR with <1% clean-accuracy drop.

**Questions:**

1. Can you provide a formal analysis of why and when the combination of forgetting events and category diversity (Component A) yields consistent ASR gains?

2. Have you tested the framework on larger datasets (e.g., ImageNet) or non-vision domains (e.g., NLP), and what challenges arise?

3. Why does Component A degrade performance under dirty-label poisoning, and how might this be addressed?

4. What are the additional runtime and resource costs of each component, and how do they scale to larger models or higher-resolution images?

5. Do you plan any human-in-the-loop studies to confirm that triggers remain imperceptible under real viewing conditions?

6. How does your modular framework compare or integrate with recent PCBAs like Combat or Narcissus?

7. Have you explored any defense or detection mechanisms (e.g., spectral analysis, data sanitization) to counteract your enhanced backdoor attacks?

**Ethical Concerns:**

["NO or VERY MINOR ethics concerns only"]

**Final Justification:**

Authors' responses have resolved most of my concerns, so I will maintain my original score (5).

**Limitations:**

Yes

**Quality:**

3

**Strengths And Weaknesses:**

- Well-structured, clear presentation; the modular design is easy to adapt.

- Component A’s dynamic balance of forgetting events and category diversity yields up to +15 pp ASR improvement over prior metrics.

- Component B’s use of perceptual metrics (GMSD) to select concealment-friendly samples is novel in clean-label attacks.

- Component C’s exploitation of RGB-channel sensitivity enhances ASR while preserving stealth.

- Experiments cover CIFAR-10/100, multiple poisoning rates, trigger types, and architectures with comprehensive ablations—yet remain limited to small-scale vision tasks.

### weakness

- The theoretical rationale for Component A’s collaboration is not formally analyzed; empirical gains lack mathematical underpinning.

- Component A performs worse under dirty-label settings (Table 6), but no remediation or analysis is provided.

- Computational overhead of the three components is not quantified, raising questions about scalability.

- Stealth is measured only via automated metrics (MSE/GMSD); no human perceptual study validates trigger invisibility.

- No concrete defenses or detection strategies are evaluated against the enhanced attack framework.

---

> ### Author Rebuttal · Authors · 2025-07-27
>
> We sincerely appreciate your valuable suggestions and will reply to all concerns in order of importance.
>
> We complete the experiments about **Backdoor Defense** and apply the components to **recent PCBAs**, achieving a **new SOTA performance** in Backdoor Attacks.  Related codes and analysis about the new results during the rebuttals will be updated in the paper.
>
> # A: Applicability on recent PCBAs and Deployment Cost
>
> {Weakness3,Question4,Question 6}
>
> ## A.1 We achieve a **new SOTA performance** based on the SOTA attack (Narcissus).
> >By poisoning merely **2 images (poison rate = 0.00004)**, Narcissus with res-log achieves 96.12% ASR and 95.10% BA in CIFAR-10 with 0 as the target-label.
>
> >We follow the default settings in the official code repository. ASRs and BAs are the mean ASR between epochs {180-200}.
>
> |Methods|ASR|BA|
> |:-|:-|:-|
> |Narcissus|46.11|94.79|
> |ours|**96.12**|**95.10**|
>
> ## A.2 We provide a **guide** to integrate all components with recent PCBAs like Narcissus and Combat.
> >Component A can be simply applied by modifying the poisoning indexes. Stronger trigger highlights the forgetting events. Triggers with larger poisoning scope and larger number of categories in the dataset highlight category diversity.
>
> >Component B selects samples by comparing the similarity before and after data poisoning, which does not require additional processing.
>
> >Recent PCBAs typically ensure stealthiness by setting a limit on pixel perturbation thresholds. Component C suffices to apply RGB differentiation processing to these thresholds when training the generator.
>
> ## A.3 Components are intended to be **flexibly applied** according to the characteristics of the trigger and task requirements.
> >For invisible attacks such as Narcissus, applying components A&C (even only A) is enough.
>
> ## A.4 Our deployment cost is **low**.
> >The cost of Component A is the same as the SOTA methods (forget), and no training is introduced in Components B or C. Component B requires only a single traversal through the target class (1/10 in CIFAR10 and 1/100 in CIFAR100) by maintaining a set with minimal metrics. Component C only requires modification of the poisoning intensity without additional overhead.
>
> # B: Lack of Defense Experiments
>
> {Weakness5,Question7}
>
> **{abl: Anti-backdoor learning,ac: activation clustering,fp: Fine-pruning, i-bau: Implicit Hypergradient, nc: Neural Cleanse, np: Reconstructive Neuron Pruning}**
>
> >Given the defense method X, bASR and X-ASR denote the pre-defense and post-defense ASR.
>
> **Badnets**
>
> |method|bASR|abl-ASR|ac-ASR|fp-ASR|i_bau-ASR|nc-ASR|rnp-ASR|
> |:-|:-|:-|:-|:-|:-|:-|:-|
> |random|18.8|0|18|14.4|8.0|18.8|10.5|
> |forget|52.9|8.4|36.3|31|17.3|52.9|8.8|
> |+A|56.2|**14**|47.7|36.5|27.9|56.2|36.3|
> |+B|37.9|3.5|25.6|20.2|**32.5**|37.9|34.9|
> |+C|53.7|5.5|28.4|28.9|7.4|1.0|24.1|
> |+A&C|**87.5**|1.6|**68**|**51.2**|14.8|**81.2**|**47.6**|
> |+B&C|54|0.2|51.8|25.9|5.9|54|0|
>
> **Blend**
>
> |method|bASR|abl-ASR|ac-ASR|fp-ASR|i_bau-ASR|nc-ASR|rnp-ASR|
> |:-|:-|:-|:-|:-|:-|:-|:-|
> |random|57.6|**15.1**|52.4|39.8|28.3|57.6|27.1|
> |forget|76.1|9.2|73.2|63.5|7|76.1|0|
> |+A|77.9|3.9|76.9|64.3|20.0|77.9|36.1|
> |+B|71.7|3.4|67.7|60.1|14.3|71.7|19.7|
> |+C|74.8|6.1|62.9|74.5|48.2|74.8|0|
> |+A&C|**97.1**|1.8|**93.9**|**98.5**|46.1|**96**|**97.3**|
> |+B&C|91|8.9|85.5|92.8|22|91|71.7|
> >The design of A&B depends on the specific requirements for concealment and ASR. The following results adequately address the concerns as the expected results of A&B are between \[B, A\].
> >Optimal improvement in ASR cannot be achieved when consideration is given to stealthiness. Therefore, components A&C exhibit the best ASR improvement.
>
> **Other attacks**.
>
> |Attack|Defense|pr|bASR(random)|ASR(random)|bASR(forget)|ASR(forget)|bASR(ours)|ASR(ours)|
> |:-|:-|:-|:-|:-|:-|:-|:-|:-|
> |ctrl|abl|0.03|91.3|66.5|95.7|57.4|96.5|**85.8**|
> |ctrl|ac|0.03|91.3|84.2|95.7|**93.6**|96.5|91|
> |ctrl|fp|0.03|91.3|94.9|95.7|97.4|97.2|**99.2**|
> |ctrl|i-bau|0.03|91.3|39|95.7|**98**|96.5|65.7|
> |ctrl|nc|0.03|91.3|1|95.7|**95.7**|94.8|94.8|
> |ctrl|rnp|0.03|91.3|26.3|95.7|42.2|96.5|**84.9**|
> |sig|ac|0.03|94.2|93.5|96.5|96.4|97.2|**97.5**|
> |sig|fp|0.03|94.2|61.6|96.5|75.8|98|**88.4**|
> |sig|i-bau|0.03|94.2|8.9|96.5|17.7|97.2|**42.4**|
> |sig|nc|0.03|94.2|94.2|96.5|96.5|98|**98**|
>
> ## B.1 Our methods **outperform** the original attacks when defended by backdoor defense methods.
>
> >Defended by nc, the ASR of CTRL drops from 91% to 1%. Optimized by our method, nc exhibits 94.8% ASR.
>
> ## B.2 The effectiveness of backdoor defenses primarily hinges on the characteristics of backdoor attacks themselves.
> >sig fails to penetrate the STRIP. In such a case, the attacks optimized by our method also remain futile.
>
> ## B.3 Our work may benefit Backdoor Defense by **considering the distinct importance of samples**.
>
> # C: Concerns about Dirty-label Poisoning
>
> {Weakness2,Question3}
>
> ## C.1 Remediation of the dirty label is **unnecessary** in our paper.
> >We aim to optimize dirty-label attacks to clean-label attacks while preserving high ASR, rendering further optimization in dirty-label scenarios less critical in our paper.
>
> ## C.2 We provide an analysis of the phenomenon.
> >Under clean-label settings with airplane as the target label, component A selects images of airplanes that least resemble airplanes based on the forgetting events with its category diversity. According to the phenomenon of models taking shortcuts discussed in another paper, models tend to rely on learning triggers as a shortcut to solve the hard task in the selected images.  Therefore, component A gets higher ASRs because generic airplane features exert minimal interference from backdoor features.
>
> >In contrast, under dirty-label settings, it is more effective to directly use cat images for poisoning instead of selecting airplane images that least resemble airplanes. The model faces the greatest difficulty in classifying cats as airplanes and resorts to backdoor shortcuts, resulting in higher ASRs.
>
> # D: Stealthiness Evaluation
>
> {Weakness4,Question5}
>
> ## D.1 **Visual presentation of poisoned images selected by Component B** with different MSD and GMSD values is provided in **Appendix I**.
>
> ## D.2 We select the advanced metric to ensure the evaluation of similarity.
> >GMSD demonstrates higher sensitivity to structural distortions (such as blurring and compression artifacts) than SSIM. Additionally, SSIM suffers from instability in white noise assessment and exhibits deviations from subjective scores under specific distortion conditions. In contrast, GMSD shows stronger consistency with human subjective evaluations.
>
> ## D.3 Your concerns are entirely justified, but this stems from inherent flaws in the evaluation metric design **rather than** flaws of Component B.
> >Component B is designed for machine-based applications. For any machine-quantifiable evaluation metric provided, it can rapidly identify images that achieve optimal performance under that metric. Given a fixed trigger, component B selects images with optimal stealthiness rather than making a visible trigger totally invisible.
>
> ## D.4 The idea of Component B is to enhance stealthiness via sample selection, which is universally applicable to both machine and human evaluations.
>
> # E: Theoretical Concerns
>
> {Weakness1,Question1}
>
> ## E.1 We relocated a significant portion of the theoretical analysis in the Appendix.
> >In total, we dedicate **Our methods** and **Appendix {B, C, D}** to establish the theoretical foundation of the components. Additionally, background knowledge of the human visual system related to Component C is provided in **Appendix E**.
>
> ## E.2 Macroscopic analysis of component A is provided in the paper.
> >Component A selects images of airplanes that least resemble airplanes based on the forgetting events with its category diversity. According to the phenomenon of models taking shortcuts discussed in another paper, models tend to rely on learning triggers as a shortcut to solve the hard task in the selected images.  Therefore, component A gets higher ASRs because generic airplane features exert minimal interference from backdoor features.
>
> >Models with triggers covering larger poisoning regions require analysis of more pixels in the corresponding areas for classification, and thus are more susceptible to being interfered with by features of other classes. Samples with larger category diversity are inherently "cross-class," which mitigates the interference.
>
> ## E.3 Model interpretability itself has long been a significant challenge in the field of AI.
> >The mathematical interpretation of Component A needs to encompass the mathematical relationship between changes in inputs (e.g., triggers and trigger variations) and corresponding changes in outputs of DNNs. A precise mathematical underpinning of the irrelation between inputs and outputs of DNNs is currently impractical.
>
> >Therefore, the mathematical sections in most papers of Backdoor Attack focus on algorithm elaboration for trigger generation and modeling of Backdoor Attack objectives.
>
> ## E.4 Our work aims to motivate the community to prioritize and investigate this new direction.
> >As a pioneering approach, establishing a comprehensive theory on the collaboration between sample selection and triggers is a long-term and arduous task.
>
> # F: Performance on Large Dataset
>
> {Question 2}
>
> ## F.1 We provide the experimental results on Tiny-ImageNet in **Appendix H**.
>
> >The ASR of the optimized Badnets is 38.96%, which is 21.90% higher than random and 6.67% higher than the SOTA metric (forget).
>
> >Most clean-label attacks exhibit ineffective performance in large datasets. For Tiny-ImageNet with 200 classes, the clean-label poisoning rate is constrained to be less than 0.005. In that case, each part of the poisoning process must be meticulously designed, thereby highlighting the value of our proposed methods.
>
> We hope our response can help resolve your confusion, and we will continue to optimize the parts that may cause confusion

---

> > ### Comment · Reviewer_CWuV · 2025-08-04
> >
> > Thank you for addressing my questions with experimental results and detailed explanations. These responses have resolved most of my concerns, so I will maintain my original score. Meanwhile, I have also reviewed the other reviewers’ comments, many of which raise valid points, and I hope you will address them thoroughly during the discussion phase. Good luck.

---

> ### Author Response · Authors · 2025-08-05
> **Thanks for the advice**
>
> We are deeply grateful for your assistance and recognition of our work. It has been a pleasure to address your concerns.

---

### Decision · Program_Chairs · 2025-09-17

**Decision:**

Accept (poster)

**Comment:**

This paper presents a modular framework comprising three generalized components that collaboratively optimize sample selection and trigger design to enhance both the attack success rate (ASR) and stealthiness of poison-only clean-label backdoor attacks. Prior to the rebuttal, reviewers generally recognized the work’s relevance and practical value, noting its novelty in jointly addressing sample selection and trigger design, as well as its use of human visual system sensitivity for trigger optimization. Nonetheless, they expressed concerns regarding its limited applicability beyond clean-label settings, insufficient evaluation against backdoor defenses, and the mixed effectiveness of component combinations. During the rebuttal and discussion periods, the authors conducted new experiments, clarified the interactions among components, and reinforced their claims through additional analyses. These responses led all reviewers to either maintain or raise their scores, resulting in a consensus leaning toward acceptance. Weighing the methodological innovation, thorough experimental validation, and constructive engagement with reviewer feedback against the remaining minor limitations, I recommend acceptance of this paper.